# Growth, Evapotranspiration, Gas Exchange and *Chl a* Fluorescence of Ipê-Rosa Seedlings at Different Levels of Water Replacement

**DOI:** 10.3390/plants13202850

**Published:** 2024-10-11

**Authors:** Kalisto Natam Carneiro Silva, Andréa Carvalho da Silva, Daniela Roberta Borella, Samuel Silva Carneiro, Leonardo Martins Moura dos Santos, Matheus Caneles Batista Jorge, Beatriz Feltrin Magosso, Mariana Pizzatto, Adilson Pacheco de Souza

**Affiliations:** 1Institute of Agrarian and Environmental Sciences, Federal University of Mato Grosso, Sinop 78550-728, MT, Brazil; kalisto.silva@sou.ufmt.br (K.N.C.S.); samuel.carneiro@sou.ufmt.br (S.S.C.); leonardo.santos@sou.ufmt.br (L.M.M.d.S.); matheus.jorge@sou.ufmt.br (M.C.B.J.); beatriz.magosso@sou.ufmt.br (B.F.M.); mariana.pizzatto@ufmt.br (M.P.); adilson.souza@ufmt.br (A.P.d.S.); 2Postgraduate Program in Agronomy, Federal University of Mato Grosso, Sinop 78550-728, MT, Brazil; 3Postgraduate Program in Environmental Sciences, Federal University of Mato Grosso, Sinop 78550-728, MT, Brazil; daniela.borella@sou.ufmt.br

**Keywords:** ipê-rosa - *Handroanthus impetiginosus* (Mart. ex DC.) Mattos, native tree, irrigation management, water sensitivity, photosynthesis, photochemical stress

## Abstract

In general, young plants in the establishment phase demonstrate sensitivity to changes in environmental conditions, especially regarding water availability. The effects of the seasonality of biophysical processes on plant physiology can trigger differential responses, even within the same region, making it necessary to conduct studies that characterize the physiological performance of the species at different spatial and temporal scales, making it possible to understand their needs and growth limits under water stress conditions. This paper aimed to evaluate the growth, gas exchange and *Chl a* fluorescence in ipê-rosa seedlings subjected to levels of water replacement (LWRs) of 100, 75, 50 and 25% in a greenhouse. The morphometric variables of plant height, diameter at stem height, numbers of leaves and leaflets, root length and volume, plant dry mass and leaf area were evaluated. The potential evapotranspiration of seedlings (ETc) was obtained using direct weighing, considering the water replacement of 100% of the mass variation between subsequent days as a reference; the cultivation coefficients (kc) were obtained using the ratio between ETc and the reference evapotranspiration (ETo) obtained by the Penman–Monteith FAO-56 method. Biomass and evapotranspiration data were combined to determine water sensitivity. Diurnal fluxes of gas exchange (net photosynthesis rate, transpiration rate, stomatal conductance, internal and atmospheric carbon ratio, water use efficiency and leaf temperature) and *Chl a* fluorescence (Fv/Fm, Φ_PSII_, ETR, Fv′/Fm′, NPQ and qL) were evaluated. Water restriction caused reductions of 90.9 and 84.7% in the increase in height and diameter of seedlings subjected to 25% water replacement when compared to seedlings with 100% water replacement. In comparison, biomass accumulation was reduced by 96.9%. The kc values increased throughout the seedling production cycle, ranging from 0.59 to 2.86. Maximum water sensitivity occurred at 50% water replacement, with Ky = 1.62. Maximum carbon assimilation rates occurred in the morning, ranging from 6.11 to 12.50 µmol m^−2^ s^−1^. Ipê-rosa seedlings regulate the physiology of growth, gas exchange and *Chl a* fluorescence depending on the amount of water available, and only 25% of the water replacement in the substrate allows the seedlings to survive.

## 1. Introduction

*Handroanthus impetiginosus* (Mart. ex DC.) Mattos, known as ipê-rosa, is a species that belongs to the Bignoniaceae family; it has a wide geographic distribution throughout the Brazilian tropical forests (Amazon Rainforest, Atlantic Forest, Caatinga, Cerrado and Pantanal) and the transitions between biomes [1,2,3]. This forest species is native to Brazil and has phenotypic and genotypic characteristics that are favorable for sustainable development in terms of environmental, economic and social aspects, which makes the conservation of this and many other forest species essential.

In the silvicultural field, *H. impetiginosus* is indicated for the recovery of degraded areas and insertion in the landscaping of urban areas [4,5,6] and in synergistic processes, such as the cycling of nutrients and carbon in natural ecosystems [7,8]. Furthermore, this tree’s importance is determined by its stem’s favorable characteristics and the average size of its trunk, with heights ranging from 8 to 30 m [1]. In addition, due to the high density of the wood (specific mass of 0.96 g cm^−3^), it is resistant to attack by insects and other pathogens, and it is used in the furniture industry and civil construction [1,2,9]. Studies claim that *H. impetiginosus* is also used in folk medicine in Brazil and other countries in South and Central America for the treatment of edema, arthritis, diuretics and infections [10,11], as the extracts and compounds isolated from its leaves have multiple pharmacological activities [12]. Furthermore, extracts and compounds based on ipê-rosa leaves are widely used to treat cancer, as they have anti-inflammatory, antioxidant and anticancer properties [13,14].

From another perspective, *H. impetiginosus* seedlings show tolerance to water deficit, a recurring environmental characteristic in degraded environments, expressing tolerance mechanisms to maintain high water potential in their tissues, such as leaf senescence and reductions in stem growth (diameter and height) and total biomass production. Furthermore, over time, they adjust their morphophysiology to stress by emitting leaves with reduced area and a greater allocation of photoassimilates to the root system [14,15].

However, the effects of the seasonality of biophysical processes on plant physiology can trigger differential responses, even within the same region [16]. In general, young plants in the establishment phase demonstrate sensitivity to changes in their growth environment; discrepancies in solar radiation levels, water availability [17,18], salinity, temperature and relative humidity of the air imprint differential responses in the plant metabolic rhythm, which can alter the photoassimilate regime (photochemical potential and biomass accumulation) [17,19,20]. Furthermore, depending on the level of environmental stress, these individuals tend to modify their biomolecular, anatomical and morphological structure beyond survival [18,21,22]. Therefore, studies that characterize the physiological performance of species at different spatiotemporal scales are essential, making it possible to understand their needs and growth limits under water stress conditions.

In this context of water availability, there is the Cerrado–Amazon transition region (transition between savanna and forest formations), which occupies an area of approximately 152,180 km^2^ and is appreciated for its wealth of natural resources, in addition to housing several important springs for the formation of large Brazilian hydrographic basins [23,24]. The resilience of the natural vegetation of this region (Cerrado–Amazon transition) to abiotic factors has sparked interest in understanding the morphophysiological behavior of native forest plants in response to stresses caused by water and salinity stress [25], irrigation cycles [15,23], nutrient availability [25,26,27], high temperatures and qualitative and quantitative variations in solar radiation [28,29,30,31], among other studies [3,32]. Despite the importance of this region of vegetation and climate transition for environmental and socioeconomic aspects, human actions without planning and adequate management of natural resources associated with climate change have generated negative impacts on these environments, such as the qualitative and quantitative loss of biodiversity and, consequently, an imbalance of plant functions in natural ecosystems. Therefore, it is necessary to carry out ecophysiological studies under abiotic stresses, such as water availability for plants, which will allow the production of information to support planning and strategic actions for the conservation of flora biodiversity in natural ecosystems.

In this sense, in this paper, we evaluated ipê-rosa seedlings in a greenhouse in the Cerrado–Amazon transition, establishing the following objectives: (i) to identify the effect of water restriction on the growth rate and final accumulation of biomass of the seedlings; (ii) to determine the accumulated crop evapotranspiration (ETc) and the cultivation coefficients (kc); (iii) to compare the water sensitivity of this species to the reduction in the amount of water available in the substrate; (iv) to verify how the gas exchange of ipê-rosa leaves responds to variations in the daily flux; (v) to determine the availability of water necessary to regulate the leaves of ipê-rosa seedlings outside the condition of photochemical stress; (vi) to verify how the *Chl a* fluorescence of ipê-rosa leaves responds to variations in the daily flux of global radiation in protected plant production environments.

## 2. Material and Methods

### 2.1. Experimental Site

The experiment was conducted for 107 days between August and December 2019, in a greenhouse in Sinop (Cerrado–Amazon transition), Mato Grosso, Brazil (11°51″51.453′ S, 55°29″6.608′ W; average altitude of 380 m) (Figure 1). This plant production structure (greenhouse) was oriented to the northeast, formed by a curved roof, with a 120-micron LDPE transparent plastic covering, and, on the sides, black polyethylene screens with 50% attenuation of solar radiation; the dimensions were 12.0 × 6.0 × 4.5 m (length, width and height), with three wooden benches inside, measuring 8.0 × 0.5 × 1.3 m (length, width and height).

According to the Köppen classification, the region’s climate is Aw, a hot and humid tropical savanna. Two water regimes are established during the year: dry (May to September) and rainy (October to April). Average monthly temperatures vary from 24 to 27 °C. Annual precipitation and reference evapotranspiration are 1970 and 1330 mm, respectively [33].

### 2.2. Obtaining and Germinating Seeds of Handroanthus Impetiginosus

The ipê-rosa pods were collected from urban afforestation matrix trees and dispersed in different points of public areas in the urban perimeter of Sinop, Mato Grosso state (Figure 1). Trees with similar dendrometric characteristics, with straight stems and globose crowns, were selected; the pods (fruits) were collected when ripe and with minimal incidence of pathogens. Under the region’s environmental conditions, this species presents the flowering and seed dispersal phenophases between July and August. After collection, the pods were processed, and the seeds with good phytosanitary aspects were stored in a natural Kraft paper bag and kept in the refrigerator (average temperature of 12 °C, relative humidity of 10%) until the beginning of the experiment.

The ipê-rosa seeds underwent aseptic treatment in a 2% sodium hypochlorite (NaClO) solution for five minutes; subsequently, they were distributed in sets of 100 units within ten transparent plastic boxes measuring 20 × 20 cm (adapted gearboxes) with Germitest^®^ paper moistened with distilled water; these seeds were sprayed with a 2% Protreat^®^ fungicide solution. For 23 days, the plastic boxes were kept in a BOD germination chamber at a constant temperature of 30 °C and a 12 h photoperiod to enable the standardization of the germination process. To ensure the germination viability of the seed batch, the number of germinated seeds (those with radicle protrusion) was counted daily for 21 days, thus allowing the determination of the following variables: germination speed index (GSI) of 10.12 ± 1.3%; average seedling height of 3.08 ± 0.87 cm; stem basal diameter average of the seedling of 1.20 ± 0.09 mm; average length of the seedling root system of 3.1 ± 1.21 cm; average volume of the seedling root system of 0.49 ± 0.09 cm^3^; total dry mass of seedlings of 0.027 ± 0.01 g.

### 2.3. Transplanting and Substrates for Seedling Production

After seedling formation (complete expansion of the pair of cotyledons leaves), a screening was performed to ensure the standardization of seedling size; subsequently, these seedlings were transplanted into 8 L plastic pots (with an effective measurement of 7.2 L of soil). The containers were filled with a mixture composed of a 2:1 ratio (forest soil/commercial substrate—the forest soil used was considered to be the surface layer (up to 0.10 m deep) of a dystrophic Red-Yellow Latosol occupied by native vegetation; it was sieved and disinfected, while the substrate consisted of a mixture of biostabilized pine bark, vermiculite, charcoal powder, water, phenolic foam and additives (fertilizers and soil improvers). Following the nutritional recommendations of Borella et al. [30] for this same substrate composition, 40, 380 and 80 g of N, P_2_O_5_ and K_2_O (in the forms of urea, simple superphosphate and potassium chloride) were incorporated per m^3^ of the substrate, respectively.

### 2.4. Substrate Moisture and Application of Treatments

Before the application of treatments related to water replacement of ipê-rosa seedlings, the water retention capacity of the substrate was determined, according to Borella et al. [30]. In this case, the masses of empty 820 cm^3^ tubes filled with the soil mixture above were determined; subsequently, the tubes were distributed in trays with water to saturate the soil through the capillary rise process, and the upper opening of the tubes was covered with plastic film to prevent water evaporation; after 48 h, the tubes were subjected to drainage of excess water, indicating that the substrate had maximum water retention capacity); then, the substrate samples were removed from the tubes, placed in aluminum foil trays, weighed (obtaining the wet weight), and kept in forced circulation ovens at 100 °C until a constant mass was obtained, to obtain the dry weight; the samples were weighed using a thousandths analytical balance (0.001 g) (BEL Engineering Ltda. company, Monza, Italy).

Furthermore, to monitor the stability of the physical–hydric characteristics of the substrate in the pots, the volumetric moisture and soil densities were obtained before the seedlings were transplanted and at 38 and 88 days after transplanting (DATs). In this case, no significant differences were observed, with the volumetric soil moisture being 0.43 ± 0.01 cm^3^ cm^−3^ and density being 0.78 ± 0.01 g cm^−3^, which resulted in a maximum amount of water retained by the substrate in the spans (7.2 L of soil) of 3.09, 3.13 and 3.18 kg. After transplanting, the pots were weighed daily to determine the potential crop evapotranspiration (ET_C_) and the need for water replacement based on the maximum water retention capacity.

From 21 DATs, the ipê-rosa seedlings were subjected to four treatments (100, 75, 50 and 25%) of level of water replacement (LWR), with four replicates. Each experimental unit (replication) consisted of eight pots, arranged in three randomized blocks inside the greenhouse; four pots with 100% LWR were weighed daily per experimental unit. The experimental steps are presented in the following flowchart (Figure 2).

### 2.5. Micrometeorological Monitoring

During the experimental period, in the environment outside the greenhouse, the daily values of average air temperature (in °C), relative humidity (in %), wind speed at 2 m height (in m s^−1^), global solar radiation (in MJ m^−2^ day^−1^), sunshine (in hours of light day^−1^) and rainfall (in mm day^−1^) were obtained at the UFMT—SINOP HOBO Meteorological Station, 10 m away from the experimental area (Sigma sensors Ltda. company, São José dos Campos, Brazil). This station is equipped with a CR 1000 data acquisition system from Campbell Scientific with the following sensors: global solar radiation (CS300 pyranometer) at 2.0 m height, psychrometer (108 Probe Temperature) at 2.0 m height and rain gauge (TE 525) at 1.50 m height, all acquired from Campbell Scientific Ltda. company, Logan City, UT, USA, and wind speed and direction (anemometer, 03002-L RM—Young Company, Traverse City, MI, USA) at 10.0 m in height. At the same time, routine, conventional insolation (Campbell–Stokes heliograph) and rainfall (rainfall height by Ville de Paris rain gauge) measurements were performed.

Inside the greenhouse, variations in air temperature and relative humidity were monitored using Instrutemp HT 4000 ICEL thermo-hygrometer dataloggers (Instrutemp, São Paulo, SP, Brazil) (operating temperature and humidity: −40 to 70 °C (−40 to 158 °F) and 0 to 100%; response time: temperature 20 s and humidity 5 s; accuracy: temperature ±1.0 °C (±2.0 °F), humidity ± 3.0% and dew point ±2.0 °C (±4.0 °F); power supply: 3.6 V rechargeable battery—lithium; and selectable measurement cycle from 2 s to 24 h), installed 1.80 m above the ground, arranged at equidistant points from the center of the structure, with readings at 30 min intervals (Instrutemp Ltda. company, São Paulo, Brazil).

To evaluate the radiation transmissivity in the protected environment (greenhouse), instantaneous readings of global radiation (H_G_) and photosynthetically active radiation (H_PAR_) were taken between 7:00 and 14:00 h, at 40 and 80 DATs. These readings were performed with MP-200 and MQ-200 pyranometers from Apogee Instruments Ltda. company, Logan City, USA, kept on a mobile and leveled metal platform; transmissivity percentages of the plastic cover of the greenhouse were obtained as 52.05 and 52.46% for H_G_ and 41.02 and 40.10% for H_PAR_, at 40 and 80 DATs, respectively. These percentages were used to obtain the global radiation inside the experimental period, considering the product with the global radiation from the meteorological station.

### 2.6. Reference Evapotranspiration, Crop Coefficients and Water Sensitivity

During the experimental period, from 28 August to 13 December 2019 (107 days), the readings of the daily crop evapotranspiration (ETc) of the ipê-rosa seedlings were taken considering the 100% level of water replacement (LWR); in the other treatments, the daily real evapotranspiration (ETr) was obtained. To obtain the daily reference evapotranspiration (ET_0_), the Penman–Monteith FAO-56 method [34] (Equation (1)) was used, based on meteorological measurements at the station. The FAO-PM method was chosen because it is recommended by the Food and Agriculture Organization of the United Nations (FAO) and combines physiological and meteorological parameters in estimating evapotranspiration [34].
(1)$${\mathrm{ET}}_{0}=\;\frac{0.408\Delta \left(\mathrm{SRD}-G\right)+\gamma \left(\frac{900}{T+273}\right)SW\left(\mathrm{es}-\mathrm{ea}\right)}{\Delta +\gamma \left(1+0.34\mathrm{WS}\right)}$$
where ET_0_ is the reference evapotranspiration (mm day^−1^); SRD is the net radiation at the surface of the crop (MJ m^−2^ day^−1^); G is the heat flux density of the soil (MJ m^−2^ day^−1^); T is the average air temperature at 2.0 m height (average value of Tmax and Tmin, °C); Ws is the wind speed at 2.0 m height (m s^−1^); es is the saturation vapor pressure (kPa); ea is the current vapor pressure (kPa); Δ is the slope of the saturation vapor pressure versus air temperature curve (kPa °C^−1^); γ is the psychometric constant (kPa °C^−1^).

In this case, the SRD was obtained based on estimates of the components of the shortwave radiation balance (considering an albedo of 0.23) and the longwave radiation balance (using the Brunt model—the insolation ratio was obtained by applying the regionally calibrated Angstrom–Prescott model [35], considering the coefficients a = 0.29 × cos[latitude] and b = 0.52). Other details of SRD, Δ and γ calculations and other parameters necessary for computing ETo were obtained according to the procedure described by Allen et al. [34]. The daily ETc and ET_0_ values ratio was used to obtain the crop coefficients (kc). Water sensitivity was evaluated through the coefficient “Ky” for seedlings [29] at 25, 50 and 75% LWRs (Equation (2)); for this purpose, they were considered as references for maximum evapotranspiration (ETm) and potential yield (Ym)—corresponding to the total dry mass of seedlings—in the 100% LWR.
(2)$$Ky=\left[\left(1-\frac{Yr}{Ym}\right)/(1-\frac{ETr}{ETm})\right]$$
where Ky = water sensitivity; Yr = actual dry mass yield (g m^−2^); Ym = maximum dry mass yield in the treatment with 100% water replacement (g m^−2^); ETr = actual accumulated evapotranspiration (mm); ETm = maximum accumulated evapotranspiration in the treatment with 100% LWR (mm, representing ET_C_).

### 2.7. Seedling Growth Analysis

Non-destructive growth analyses were performed from 28/09/2019 to 13/12/2019, corresponding to 31, 59, 71, 87 and 107 days after transplanting (DATs); all plants in each experimental unit were measured in each evaluation. On these occasions, the variables analyzed were as follows: (i) seedling height—measured from the plant basal to the insertion of the apical bud with the aid of a graduated ruler (0.1 cm); (ii) basal stem diameter (BSD)—obtained at the height of the substrate with the aid of a centesimal digital caliper (0.01 mm); (iii) number of leaves; (iv) number of leaflets per seedling.

After the end of the experimental period, at 107 DATs, destructive analysis of the plants mentioned above was performed to obtain the following: (i) leaf area in cm^2^—with a photoelectric meter (LI-3100C) (LI-COR Ltda. company, Lincoln City, OR, USA); (ii) root length—with the aid of a graduated ruler (0.1 cm); (iii) root volume—obtained with the aid of a graduated cylinder (0.1 mL), by immersing the washed roots in a known volume of water; (iv) dry matter masses of roots (DMR), stems (DMS), leaves (DML) and total (sum of partitions)—determined on an analytical balance with an accuracy of 0.0001 g (Mettler-Toledo Ltda. company, Barueri, Brazil) after the plant material was dried in a forced air circulation oven at a temperature of 65 °C ± 5 °C until a constant mass was obtained.

### 2.8. Gas Exchange and Fluorescence

The gas exchange evaluations in the leaves of ipê-rosa seedlings were performed on only one date, 100 days after transplanting, using a portable photosynthesis analyzer (IRGA), model LCi-SD from ADC BioScientific Ltda. company, Hoddesdon, UK. Three plants were selected for each treatment of 25, 50, 75 and 100% water replacement, and three readings were taken from each treatment using the median portion of the second or third adult leaf from the apical bud.

The readings were taken throughout the day at the following solar times: 09:30, 10:30 and 11:30 (a.m.); 12:30, 13:30, 14:30 and 16:00 h (p.m.). The leaves were adjusted inside the IRGA analyzer chamber, occupying an area of 6.25 cm^2^. They were subjected to an effective pulse of 1839 μmol m^−2^ s^−1^ of light intensity to obtain the following variables: net photosynthesis rate—A (μmol m^−2^ s^−1^); transpiration rate—E (mol m^−2^ s^−1^); stomatal conductance—gs (mol m^−2^ s^−1^); internal carbon concentration—Ci (μmol mol^−1^); carbon concentration in the external environment—Cref (μmol mol^−1^); carbon variation between the external environment and analysis chamber—ΔC (μmol mol^−1^); leaf temperature—TF (°C). Water use efficiency, EUA (μmol CO_2_ mol^−1^ H_2_O), was obtained using the ratio between A and E; the relationship between internal carbon and atmospheric carbon—Ci/Ca—was given by the ratio between Ci and the difference between Cref and ΔC.

The fluorescence of *Chl a* in leaves of ipê-rosa seedlings was determined with an Os5p modulated light fluorometer (Brand: Opti-Sciences Ltda, Hudson City, NY, USA, Fluorpen, Model: FP-100). The protocols for the analyses were the Fv/Fm Protocol (Fv/Fm Protocol, measurements in the dark-adapted state) and Yield Protocol (Yield Protocol, measurements in the light-adapted state). The fluorescence evaluations occurred at 28, 66 and 90 DATs, for 5 plants per experimental unit, regardless of the treatment. The second or third adult leaf from the apical bud on the respective date was used to perform the readings.

*Chl a* fluorescence reading before dawn (predawn) was performed between 4:30 and 5:10 h using the Fv/Fm Protocol to obtain the photochemical performance of PSII after the complete regeneration of protein complexes during the night, as recommended by Elsheery and Cao [36]. The fiber optic bundle was placed against the leaf tissue using plastic clips to emit a pulse of modulated light incapable of triggering photochemical chains (0.05 μmol m^−2^ s^−1^) to measure the minimum fluorescence before dawn (Fo_predawn_). At an interval of approximately 0.8 s, a pulse of saturating actinian light with an intensity greater than 5000 μmol m^−2^ s^−1^ was emitted to measure the maximum fluorescence before dawn (Fm_predawn_). The interval between these two points provided the variable predawn fluorescence (Fv_predawn_), used in calculating the maximum quantum yield of PSII before dawn (Fv/Fm_predawn_), according to Equation (3).
(3)$$Fv/Fm=\frac{Fm-Fo}{Fm}$$
where Fv/Fm = maximum quantum yield of PSII; Fm = maximum fluorescence in the dark-adapted state; Fo = minimum fluorescence in the dark-adapted state.

After that, *Chl a* fluorescence analyses were performed throughout the day. The first assessment was performed between 6:30 a.m. and 5:30 p.m. at sequential one-hour intervals. In the two subsequent assessments, readings were interrupted at 1:30 p.m. These analyses, on the same plants and leaves as the predawn sampling, were performed according to the equipment protocols: dark-adapted and light-adapted states.

For the “dark-adapted state” protocol, the leaves were acclimated to the dark for 30 min. After the interval, the Fv/Fm Protocol was repeated using the same procedures described for predawn. Thus, using Equation (3), the maximum quantum yield of PSII (Fv/Fm) was determined on the hourly scale for the treatments in which each plant was a replicate.

The Yield Protocol was used in the “room light-adapted state” condition. For this purpose, the leaf tissue was exposed to a pulse of actinia light with a photosynthetically active photon flux (FFF) of approximately 560 μmol m^−2^ s^−1^ for 5 s to obtain the variable F′ (fluorescence in the light-adapted state). Subsequently, the leaf was subjected to progressive saturating actinia light pulses (5000, 10,000 and 15,000 μmol m^−2^ s^−1^), estimating the maximum fluorescence of the light-adapted state (Fm′) in a curve at infinity. Through this projection, the fluorometer provided the effective quantum yield value of PSII (Φ_PSII_), according to Equation (4).
(4)$$\Phi_{PSII}=\frac{Fm^{\prime }-F^{\prime }}{Fm^{\prime }}$$
where Φ_PSII_ = effective quantum yield of PSII; F′ = fluorescence in the light-adapted state; Fm′ = maximum fluorescence in the light-adapted state.

For fluorescence in the light-adapted state, three readings were taken on each leaf per plant at each hour of analysis. Using the variables above and the equations described by Oxborough and Baker [37], Lichtenthaler et al. [38] and Baker [39], the other variables related to Chl a fluorescence were estimated. The minimum fluorescence in the light-adapted state (Fo′), considering the complete oxidation of QA, in this case, through the transfer of electrons from PSII to PSI, was obtained according to Equation (5).
(5)$$Fo^{\prime }=\frac{Fo}{\left(Fv/Fm)+(Fo/Fm^{\prime }\right)}$$
where Fo′ = minimum fluorescence in the light-adapted state; Fo = minimum fluorescence in the dark-adapted state; Fv = variable fluorescence in the dark-adapted state (difference between Fm and Fo); Fm = maximum fluorescence in the dark-adapted state; Fm′ = maximum fluorescence in the light-adapted state.

Therefore, the following parameters were also obtained: (i) the maximum efficiency of PSII (Fv′/Fm′), which provides the potential capacity of PSII, with tissues acclimated to light conditions (Equation (6)); (ii) non-photochemical dissipation (NPQ), which characterizes the portion of photochemical energy extinguished by heat loss (Equation (7)); (iii) the open fraction of PSII reaction centers (qL), which provides the capacity of the photochemical apparatus to receive electrons considering the complete oxidation state of QA (Equation (8)).
(6)$$Fv^{\prime }/Fm^{\prime }=\frac{Fm^{\prime }-Fo^{\prime }}{Fm^{\prime }}$$
where Fv′/Fm′ = maximum efficiency of PSII; Fm′ = maximum fluorescence in the light-adapted state; Fo′ = minimum fluorescence in the light-adapted state.
(7)$$NPQ=\frac{Fm-Fm^{\prime }}{Fm}$$
where NPQ = non-photochemical quenching; Fm = maximum fluorescence in the dark-adapted state; Fm′ = maximum fluorescence in the light-adapted state.
(8)$$qL=\frac{\Delta F}{\left(Fm^{\prime }-Fo^{\prime }\right)}\times \frac{Fo^{\prime }}{F^{\prime }}$$
where qL = open fraction of PSII reaction centers; ΔF = difference between Fm′ and F′; Fm′ = maximum fluorescence in the light-adapted state; Fo′ = minimum fluorescence in the light-adapted state; F′ = fluorescence in the light-adapted state.

Finally, the electron transport rate (ETR) was estimated based on the photosynthetically active photon flux (FFF) of 560 μmol m^−2^ s^−1^, which provides the portion of the incident flux that is effectively carried to photochemical activity in the electron transport chain, according to Equation (9).
(9)$$ETR=0.84\times 0.50\times FFF\times \Phi_{PSII}$$
where ETR = electron transport rate (μmol m^−2^ s^−1^); FFF = photosynthetically active photon flux of 560 μmol m^−2^ s^−1^; ΦPSII = effective quantum yield of PSII.

Concurrent with the readings with the fluorometer and portable photosynthesis analyzer (IRGA), instantaneous measurements of global radiation (with MP-200), photosynthetically active radiation (with MQ-200) and luminance (Instrutherm lux meter LD200) were performed.

### 2.9. Basal Temperatures and Thermal Sum

To evaluate the variables studied in this study, the thermal sum, in accumulated degree days (DDAs), was adopted as the independent variable to ensure greater applicability of the results to other regions where ipê-rosa occurs. To this end, estimates of the upper (TB) and lower (Tb) basal temperatures were made using the database belonging to the Plant and Environment Interactions Research Group, collected by Monteiro [40] for seedlings of this same species subjected to different shading conditions in the region’s dry season (April to August); in this case, the ipê-rosa seedlings were produced in full sunlight conditions, covered with polyethylene screens in the colors blue, red and green with respective percentages of 53, 57 and 57% of attenuation of global radiation and black screens with 38, 52 and 83% of attenuation; these shading conditions generated different microclimates that interfered with seedling growth, allowing basal temperatures to be adjusted by relationships with leaf area and dry mass. Using daily data on maximum and minimum air temperatures throughout the experimental period and in each cultivation condition, TB and Tb values were obtained by polynomial regressions (second degree), whose independent variables (x) were, respectively, maximum and minimum air temperatures of each treatment (shading screens), associated in isolation with the dependent variables total average leaf area per seedling (y1) and total average dry mass (y2), resulting in four polynomial adjustments (Figure 3). The first derivative of the adjusted equations was used to establish TB and Tb, obtaining the respective maximum points of the functions and their corresponding TB and Tb estimated at 40.6 and 16.9 °C, respectively. After TB and Tb were defined, the thermal sum of the ipê-rosa seedlings in this study was determined using the methodology of Ometto [41], according to the recommendations of Souza et al. [42].

### 2.10. Data Analysis

The data were subjected to a normality test and subsequently to an analysis of variance; when significant, the means were compared by the Tukey test at 5% probability in the Sisvar 5.6 software [43], with graphical guidelines and polynomial adjustments performed in the Origin 2018 software. The analyses of the growth, evapotranspiration, gas exchange and fluorescence variables were performed based on the accumulated thermal sum, to minimize the effects of the production environment on the growth rates and cell expansion of the plant.

To evaluate the non-destructive growth variables (height, diameter at the height of the stem, number of leaves and number of leaflets), to adjust the significant polynomials resulting from the regression analysis, the accumulated thermal sum at 31, 59, 71, 87 and 107 DATs corresponded to 508, 935, 1125, 1355 and 1655 DDAs, respectively.

Data were collected at 107 DATs (1655 DDAs) to evaluate the destructive growth variables (root length, root volume, leaf area and total dry mass). They were subjected to an analysis of variance (F test). If significant differences were found, the means were compared using the Tukey test at a 5% probability level.

The reference evapotranspiration and the crop under 100% water replacement conditions were analyzed in a grouped (cumulative) and isolated manner as a function of the thermal sum of the ipê-rosa seedlings in accumulated degree days (DDAs) on the dates of the above-mentioned non-destructive growth analyses. For evapotranspiration at 25, 50, and 75% LWRs, the data were analyzed only in a grouped manner, as well as a function of the thermal accumulation of the plants under ETc conditions. The crop coefficients (kc) were obtained daily and subjected to simple linear regression adjustments with the thermal sum as the independent variable. No applied statistical tests were performed regarding the water sensitivity coefficients (Ky).

For gas exchange, a completely randomized design (CRD) was implemented on an hourly scale, with measurements at 09:30, 10:30, 11:30, 12:30, 13:30, 14:30 and 16:00 h. Finally, the gas exchange variables above, mean air temperature and dPv (water vapor pressure deficit) were subjected to Pearson correlation analysis (r) separately within each fluid replacement level; the coefficients of these correlations were analyzed for significance using the T-test at 1 and 5% probability levels. *Chl a* fluorescence analyses were performed at 28, 66 and 90 DATs, corresponding to 464, 1046 and 1400 DDAs, respectively. For the predawn evaluation of *Chl* a fluorescence, a CRD design (completely randomized, with 5 replicates per experimental unit) was adopted. The analyses of *Chl* a fluorescence during the day is arranged in graphical form, with the average values obtained every two hours. Therefore, the results of 06:30 and 07:30 h were grouped in the averages of 07:00, the following averages of 09:00 h were obtained with the results of 08:30 and 09:30 h, and so on. Thus, in the first evaluation, at 28 DATs (accumulation of 463.6 DDAs), the average values were ordered in the following time scale: 07:00, 09:00, 11:00, 13:00, 15:00 and 17:00 h, while for the other dates, 66 and 90 DATs (accumulations of 1046.1 and 1400.0 DDAs, respectively), the points stopped at 13:00 h.

Only for the diurnal flows of the evaluations at 66 and 90 DATs, analogous to gas exchange, the *Chl* a fluorescence, mean air temperature and dPv variables were also subjected to Pearson correlation analysis (r). Furthermore, the Fv/Fm information was extracted to establish the simple linear regression adjustments as a function of the mean air temperature and dPv, following the recommendations of Marenco et al. [44]. For this purpose, Fv/Fm was the dependent variable (y) while, in isolation, mean air temperature and dPv were the independent variables (x). Eight equations were evaluated, given the division into two independent variables—TMair and dPv—and four water replacement levels, considering significance by analysis of variance (F test) at the 1 and 5% probability levels.

## 3. Results

### 3.1. Microclimate Dynamics Throughout the Experimental Period

The microclimatic dynamics throughout the experimental period in the greenhouse are arranged according to the accumulated thermal sum of the ipê-rosa seedlings (Figure 4), starting with transplanting (28 August 2019) and the last destructive growth analysis (13 December 2019), equivalent to 0.0 and 1655 accumulated degree days (DDAs). The experiment was carried out during the dry–rainy and rainy transition seasons of the mid-northern region of Mato Grosso (between August and December). During this period, the average values of temperature and relative humidity inside the greenhouse were 29.19 °C and 66.76%, with minimums and maximums of 18.60 to 47.00 °C and 16.00 to 98.60%, respectively (Figure 4A,B).

Inside the greenhouse, global radiation ranged from 0.95 to 11.81 MJ m^−2^ day^−1^, while in the outdoor environment (weather station), the variations were 1.83 and 24.52 MJ m^−2^ day^−1^. There was an accumulation of 438 mm of rainfall, concentrated in the final period of the experiment. The plastic covering of the greenhouse does not allow rainfall to occur inside the facility; however, variations in air temperature, relative humidity and internal global radiation show relationships with the influence of water vapor and cloudiness. At the beginning of the rainfall (at 29 DATs), inside the greenhouse, there was a reduction in the daily thermal amplitude of 4.95 °C, in addition a decrease of 2.11 °C in the average air temperature (Figure 4A); in the average daily relative humidity, there was an increase of 24.7% after the beginning of the rains (Figure 4B).

### 3.2. Growth of Ipê-Rosa Seedlings

The combination of water replacement levels and accumulated thermal sum (associated with the time after transplanting—DATs) significantly influenced the increase in height, diameter at stem height (DSH) and numbers of leaves and leaflets of ipê-rosa seedlings in the greenhouse (Figure 5). These findings have practical implications for seedling production, inspiring us to optimize our methods and improve the quality of our seedlings. Based on the adjusted equations (regressions), at the end of the cycle (thermal accumulation of 1655 DDAs (107 DATs)), the respective heights of 10.25, 30.43, 45.45 and 53.12 cm were estimated for ipê-rosa seedlings subjected to 25, 50, 75 and 100% LWRs; as for the diameter at stem height (DSH), in the same thermal sum, the values of 4.05, 11.24, 14.52 and 16.17 mm were obtained, at the abovementioned levels of water replacement.

These variables (seedling height and DAC) in all water levels presented adjustments by increasing linear equations (Figure 5 and Appendix A). The growth rates decreased with the increase in water restriction, evidenced by the angular coefficients (B1) of the regressions presented in Appendix A. Based on these regressions, at 100% LWR, the seedlings presented growth rates of 0.0441 cm °C^−1^ and 0.0124 mm °C^−1^ for height and DAC, respectively, whereas, with 25% LWR, the seedlings already showed rates of 0.0040 cm °C^−1^ and 0.0019 mm °C^−1^ for the same variables, resulting in reductions of 90.9 and 84.7% in the growth rate for each accumulated degree day.

The numbers of leaves and leaflets showed different behaviors depending on the thermal sum at different LWRs (Figure 5C,D). The seedlings subjected to 50 and 75% LWRs in the leaf count showed a linear increase. At 100% and 25% LWRs, the regressions of these morphometric variables showed quadratic behaviors with concavities downwards (number of leaves) and upwards (number of leaflets).

The gradual distance between the LWR responses led to mean differences between the curves of 25–50%, 50–75%, and 75–100% of 8.69, 6.46 and 3.29 cm for height, 3.07, 1.40 and 0.71 mm for diameter at stem height and 8.25, 6.40 and 3.89 for the number of leaflets, respectively (Figure 5A,B,D).

The leaf area and the allocation of aerial and root biomass were influenced by the levels of water replacement at the end of the production cycle of ipê-rosa seedlings [at 107 DATs, with an accumulation of 1655 DDAs] (Figure 6).

Among seedlings subjected to 100% and 25% LWRs, 54.4, 92.5, 95.2 and 96.9% reductions were observed in root length, root volume, leaf area and total dry mass, respectively.

The ipê-rosa seedlings showed average percentages of 56.6, 26.7 and 16.8% for the allocation of carbohydrates in the dry mass of leaves, stems and roots, respectively, with variations of up to 9.06% between the levels of water replacement (Figure 6).

### 3.3. Evapotranspiration, Crop Coefficients and Water Sensitivity (Ky)

Daily evapotranspiration oscillated outside the microclimatic behavior shown in Figure 4. Before the onset of rainfall, at 24 DATs (398 DDAs), daily evapotranspiration of ipê-rosa seedlings (ET_C_) varied between 2.55 and 6.79 mm day^−1^; after this period, ETc oscillated between 1.64 and 9.89 mm day^−1^; similar changes were also observed in reference evapotranspiration (ET_0_) before and after the onset of rainfall (Figure 7A).

Young ipê-rosa plants subjected to 100% LWR accumulated 593.2 mm of ETc over 1655 DDAs; this water depth was 68.84% above the ETo in the same period—362.1 mm (Figure 7B). Until 33 DATs (542 DDAs), the ETc and ET_0_ curves were close to each other; however, there was a gradual separation of ETc due to leaf expansion. At a water replacement level of 75%, the accumulation of water depth was 28.5% of ET_0_, with 465.4 mm at 107 DATs. However, it was only after 68 DATs that the accumulated water replacement depth in this treatment was higher than the accumulation of ETo. The 25 and 50% LWR presented accumulated water depths of 213 and 346.2 mm at 107 DATs.

The cultivation coefficients (kc) of ipê-rosa seedlings in a protected environment were represented by an increasing linear regression dependent on accumulated degree days; in this sense, the minimum and maximum coefficients were estimated at 0.59 and 2.86 for the beginning (0 DATs—0 DDAs) and the end (107 DATs—1655 DDAs), respectively. The average kc throughout the seedling cycle (107 DATs) was 1.75 (Figure 8). This average value is higher than that described by Monteiro et al. [28], who, when evaluating seedlings of this species kept in polyethylene bags with a volume of 4.7 L, obtained a gradual reduction in kc according to the increase in shading levels, with a kc of 1.03 in the condition of full exposure to the sun and of 0.54 under 80% attenuation of solar radiation.

The averages of the parameters that make up the water sensitivity coefficient (Ky) are shown in Table 1. The dry mass yield was obtained for each level of water replacement, with the potential yield based on seedlings subjected to 100% water replacement. The Ky value represents the degree of tolerance of plants to water stress conditions; when this index is low, it demonstrates the resistance of the seedlings to the lack of water, and in the opposite case, the increase in Ky implies a greater degree of sensitivity of the seedlings to the irrigation deficit [29]. By default, the biomass yield tends to decrease as the intensity of the water restriction increases; thus, the sensitivity to water stress is classified into four categories: Ky < 0.85—low; 0.85 < Ky < 1.00—low/medium; 1.00 < Ky < 1.15—medium/high; and Ky > 1.15—high [29].

### 3.4. Diurnal Variations in Gas Exchange in Ipê-Rosa Seedlings

At 100 DATs (1558 DDAs), gas exchange assessments of ipê-rosa seedlings were carried out inside the greenhouse. Regarding the micrometeorological behavior of that day, significant differences were observed between the morning and afternoon periods for the hourly averages of the mean air temperature, dPv and PAR; in this case, the maximum values were 42.9 °C, 5.4 kPa and 937.6 µmol m^−2^ s^−1^, respectively (Appendix A).

This micrometeorological dynamic influenced the gas exchange of ipê-rosa seedlings (Table 2). The net photosynthesis rates (A) of seedlings at 25, 50 and 75% LWRs showed maximums at 9:00 h, being 6.11, 12.50 and 11.12 µmol m^−2^ s^−1^, respectively. However, for the leaves of seedlings subjected to 100% LWR (reference condition), the maximum A values (10.41 µmol m^−2^ s^−1^) were recorded at 12:30 h; the minimum A values were observed in the afternoon period regardless of the water replacement level. Water use efficiency (WUE) and stomatal conductance (gs) showed lower values close to solar noon (smaller zenith angle and higher incidence of global radiation).

The ratio between internal carbon and atmospheric carbon (Ci/Ca) of ipê-rosa leaves was above 0.50 at all levels of water replacement, reaching 0.78 at 2:30 p.m. for seedlings subjected to 100% LWR; in this case, the lowest values were also obtained at times close to solar noon. However, the degree of stomatal opening remained regular at 12:30 p.m. compared to other times and was concomitant with the highest A values for seedlings with 100% LWR.

Gas exchanges were influenced by the interaction between water replacement levels and analysis times (Table 2). The leaves of seedlings subjected to 25% water replacement showed a lower net photosynthesis rate than the other treatments at the four times evaluated. In addition, it was also noted that at 25 and 100% water replacement, no differences were observed in the A value throughout the day; however, assimilation was enhanced during the morning period. Stomatal conductance showed patterns similar to the average values of A and E. In contrast, the EUA and temperatures of the leaves of ipê-rosa seedlings did not differ between the water replacement levels. Due to the microclimatic characteristics of the growth environment at 100 DATs, the analyzed gas exchange variables were associated with each other and with the mean air temperature and dPv in a distinct manner between the water replacement levels, as can be observed by the Pearson correlation coefficients (r) with value coefficients and their degrees of significance (Table 3).

The leaves acclimated to intermediate water levels, 50 and 75% LWRs, presented the highest significant correlations, while Ci/Ca presented the lowest. Regardless of the water replacement, leaf temperature stood out for correlating with the highest number of variables (except for “gs”), and, of these, the correlations were negative only with WUE and Ci/Ca (Table 3). The highest correlation coefficients occurred for the interactions between A, E and gs (all above 0.70). However, when confronted with the characteristics of the growth environment (T_M_ and dPv), these variables showed little linearity, with 37.5% of the coefficients not having any degree of significance and the “r” values not exceeding 0.67 (r for E and T_M_ at 50% LWR).

### 3.5. Chl a Fluorescence: Seasonality and Diurnal Variations

The *predawn Chl a* fluorescence of ipê-rosa plants was affected by water replacement (WR) levels and the time of the seedling cycle (associated with DDAs). At 28 DATs (464 DDAs), the maximum photochemical quantum yield of PSII before dawn (Fv/Fm_predawn_) was similar among treatments, with an average of 0.80. In later evaluations (1046 and 1400 DDAs), only in plants maintained at the 25% LWR was a significant decrease in Fv/Fm_predawn_ observed at 0.73 and 0.76, respectively (Figure 9).

The decrease in Fv/Fm_predawn_ values of plants subjected to 25% LWR with increasing experimental period and accumulation of 936 DDAs (72 DATs, time between the first and last analysis of *Chl a* fluorescence) was 5.30%. Similar dynamics were observed for minimum (Fo_predawn_) and maximum (Fm_predawn_) fluorescence before dawn, with the same treatment patterns. With development, plants grown at 50, 75 and 100% LWRs showed decreases in Fo_predawn_, values of 180 (reduction of 31.56%) and 194 (reduction of 26.24%) in the second and third evaluations, respectively; those maintained at 25% LWR remained with a high value (255). However, the last analysis showed a relative reduction of 11.41% to 233, associated with the thermal sum of 1400 DDAs (Figure 9A). For all water treatments, the highest values of maximum predawn fluorescence (Fm_predawn_) occurred at 28 DATs, when the plants accumulated 464 DDAs, an average of 1343, with a generalized decrease in subsequent evaluations (Figure 9B). However, only at 90 DATs (thermal sum of 1400 DDAs) was a difference recorded in Fm_predawn_ for the water treatments, restricted to the lower level of 25% LWR compared to the others, which remained equal among themselves.

The internal diurnal micrometeorological fluctuations of the greenhouse, which occurred in the seedling growth environment in the three evaluations above, indicated thermal amplitudes of 12.02, 10.35 and 6.22 °C and dPv with variations of 4.12, 3.59 and 1.64 kPa for the evaluations at 464, 1046 and 1400 DDAs, respectively. On these days, the insolation was 7.1, 7.4 and 9.9 h, with a maximum daily PAR radiation of around 1016 µmol m^−2^ s^−1^), at 11:00h solar time. At 28 DATs (464 DDAs), there was a similarity in the maximum quantum yield of PSII (Fv/Fm) at the water replacement levels (Table 4), with average differences of 0.42%. In the second evaluation, when the plants had accumulated 1046 DDAs, similar Fv/Fm averages were recorded until 09:00h, except for the seedlings subjected to 25% LWR. The results were similar to those of the plants with the minimum water replacement (Fv/Fm of 0.72), which maintained low values as the seedlings developed. Due to the favorable condition of the micrometeorological components in the last evaluation (1400 DDAs) (Appendix A), mainly due to the air temperature and dPv levels below the respective monthly extremes, the Fv/Fm values were stable in all treatments during the day, with an average variation of 1.24%. In this case, the only thing that stands out is the inferiority of the plants at 25% LWR compared to the overlap of the other curves (Table 4).

Table 4 also presents the average values of the diurnal variations of *Chl a* fluorescence in the light-adapted state. When compared with the average values of the Fv/Fm variable, there was less ordering of the diurnal fluxes in the water replacement levels on the three dates analyzed for the effective quantum yield of PSII (Φ_PSII_), electron transport rate (ETR); maximum efficiency of PSII (Fv′/Fm′) and non-photochemical dissipation (NPQ), characterized mainly by inversions and overlaps between the average points in the water treatments. Nevertheless, similarly to the Fv/Fm variable, at 28 DATs, when plant development reached the accumulation of 464 DDAs, *Chl a* fluorescence in the light-adapted state of the ipê-rosa seedlings showed little distinction between the water regimes evaluated (Table 4). Furthermore, in the following evaluations, there was a distancing of the responses of the ipê-rosa seedlings subjected to 25% LWR in comparison to the other treatments, which, in turn, maintained a certain proximity during the day. A priori, although the amount of water available did not influence the current operation of the PSII, at 11:00 and 13:00 h, more accentuated favorable responses were identified in the plants at 100% LWR compared to the others, with a relative superiority of 7.59, 7.59 and 9.00% for Φ_PSII_, ETR and Fv′/Fm′, respectively, and relative inferiority of 22.92% for NPQ; however, at the next moment (15:00 h), the curves met again.

In the second evaluation, Φ_PSII_ and ETR showed the maximum point in the early morning at 25, 50 and 75% LWRs, respectively, 0.19, 0.44 and 0.50 for Φ_PSII_ and 44.32, 104.86 and 118.90 µmol m^−2^ s^−1^ for ETR, with a subsequent decrease at 09:00 h. Although the individuals at 100% LWR demonstrated behavior equivalent to that described, their maximum points occurred only after the decay between 9:00 and 11:00 h, at 13:00 h, with Φ_PSII_ of 0.51 and ETR of 121.21 µmol m^−2^ s^−1^ (Table 4).

According to Table 4, in the last evaluation with the accumulation of 1400 DDAs, within the hourly course, the dynamics of ΦPSII and ETR were approximately linear, not regular, decreasing for the seedlings maintained at 25% LWR and increasing in those with 50, 75 and 100% LWRs. On that date, despite the decreasing Φ_PSII_ and ETR values for the seedlings under water stress, 25% LWR, the results were superior to those obtained by these plants when accumulating 1046 DDAs, an average percentage of 88.4% above, yet below the other treatments. Only when the development of the seedlings reached the accumulation of 1046 DDAs, a behavioral pattern was observed in the Fv′/Fm′ curves, an asymmetric parabola with an upward concavity, with maximum values at 7:00 a.m. and minimum values close to noon local time (Table 4).

On the other dates, the ipê-rosa seedlings subjected to 100% LWR showed multiple inverse Fv′/Fm′ trends in the other measured flows when the average values were compared to the other treatments. At 90 DATs, with the accumulation of 1400 DDAs, for example, while individuals at 25, 50 and 75% of RA expressed a generalized decrease in response to the sudden gain of PAR at 13:00 h (Table 4), those subjected to full water replacement presented a maximum mean value (0.70) within the three occasions studied (Table 4). At 66 DATs, with the accumulation of 1046 DDAs, non-photochemical dissipation (NPQ) in the leaves of ipê-rosa seedlings presented a performance inverse to ΦPSII. The maximum values occurred at 09:00 h for the leaves of individuals at 50, 75 and 100% of LWR, with averages of 2.77, 2.27 and 2.07, respectively, while for the leaves growing at 25% LWR, the maximum value was concomitant with the moment of maximum daily PAR, at 11:00 h, an average of 3.39 (Table 4). However, in the third evaluation, 90 DATs, the maximums and minimums were dispersed throughout the evaluated courses in the treatments, with low comparability to Φ_PSII_; the inverse pattern, in this case, an increasing linear trend, was observed only for the 25% LWR flow (Table 4).

The seedlings acclimated to 25% LWR demonstrated a lower percentage range for NPQ, with a variation relative to the average of daily courses of 25.4 and 43.8% at 66 and 90 DATs, with accumulations of 1046 and 1400 DDAs, respectively. In contrast, the other levels of water replacement maintained a percentage range between 40.1% (100% LWR at the thermal sum of 1046 DDAs) and 97.3% (75% LWR also at the thermal sum of 1046 DDAs). Regarding the diurnal fluxes of the fraction of open reaction centers of PSII (qL), in the first evaluation, there was little differentiation between the leaves of the ipê-rosa seedlings in the LWR. In this case, with the occurrence of the maximum point at 07:00 h (average of 0.68), there was a subsequent decrease and stability during the hours, until an abrupt reduction to the minimum point (average of 0.22) at 17:30 h (Table 4). At 28 DATs, with the plants accumulating 464 DDAs, at the end of the day, the regeneration of the photochemical capacity of PSII was observed, arranged by Fv/Fm and Fv′/Fm′, along with a reduction in the effective PSII activity according to Φ_PSII_ and ETR. This qL dynamic was not repeated in the other evaluations, in which the minimum points were concentrated in the early morning, between 7:00 and 9:00 h, in general, with an increasing trend until the cessation of readings at 13:00 h (Table 4).

*Chl a* fluorescence and microclimatic components—mean air temperature and dPv—showed distinct comparability in water replacement levels with the grouping of readings at 66 and 90 DATs, with accumulations of 1046 and 1400 DDAs, dates when the plants were already acclimated to water availability. Measurements were given based on Pearson’s correlation coefficients (r), which showed different significance indices and values (Table 5). The maximum efficiency of PSII (Fv′/Fm′) showed significant “r” coefficients in almost all interactions analyzed, except with qL at 25% LWR, fluctuating in modular values from 0.55 to 0.91, positively associated with Fv/Fm, Φ_PSII_ and ETR. On the other hand, it was negatively linked to mean air temperature, dPv, NPQ and qL. Excluding the non-significant interaction coefficient (qL), the highest r value of the relationship with Fv′/Fm′ occurred with the minimum water supply level, a modular mean of 0.83, followed by 0.77, 0.74 and 0.71 at 50, 75 and 100% LWRs, respectively.

Non-photochemical dissipation (NPQ) showed a similar significance pattern to Fv′/Fm′. However, it showed more unsatisfactory coefficients, mainly regarding mean air temperature, dpv, Fv/Fm and qL, excluding the responses for the leaves of individuals at 75% LWR, whose coefficients were mostly above 0.50. An inversion was observed in the interactions of Φ_PSII_ and qL with mean air temperature and dPv in the treatments. The correlation coefficients between qL and the microclimate were significant for the leaves of ipê-rosa plants subjected to 50, 75 and 100% LWRs, which did not occur for the plants at 25% LWR. In the interaction of ΦPSII with the microclimate, the opposite occurred to what was described; however, in this case, only at the 5% probability level, the r value was also significant at 50% LWR. As expected, the ETR values matched the Φ_PSII_ results.

As an alternative to Fv′/Fm′, the Pearson correlations of the maximum quantum yield of PSII (Fv/Fm) with the average air temperature and dPv were, in the module, decreasing in the water treatments, with the maximum average for 100% LWR (−0.89) and minimum at 25% LWR (−0.72). The interactions between the maximum quantum yield of PSII (Fv/Fm) and the microclimatic variables were translated into linear regression fits at each LWR, as shown in Table 5.

## 4. Discussion

### 4.1. Morphometric Growth of Ipê-Rosa Seedlings

During the experimental period, global radiation and air temperature variations related to cloud cover and seasonal rainfall in the region were observed [45]. The dynamics of the climatic elements recorded during the experimental period result from the region’s water seasonality, independent of the degree of shading resulting from the protected environments [29,46,47].

The application of treatments associated with levels of water replacement (LWRs) began at 21 DATs when the seedlings had already accumulated 345 DDAs. This timing is crucial, as it allows us to experiment in a protected seedling production environment, minimizing the effects of rainfall. Such effects could potentially homogenize the humidity of the tube substrates and thus minimize the effects of water replacement treatments. The period of execution of the experiment was carefully chosen to align with the seed dispersal season of the species and the production systems of native and exotic seedlings in Brazilian forest nurseries. These facilities have a planning basis for producing and hardening seedlings during the transition from dry to rainy seasons, ensuring higher survival rates in the field.

The adjustments to the mathematical models of the growth variables as a function of the thermal sum served a crucial purpose in our experiment. They allow us to predict DDA values so that the forest seedlings reach the ideal morphometric level to be taken to the field. These models consider the seasonality of air temperature within the production cycle and the role of this micrometeorological variable in plant physiology and cycle length, which does not necessarily include seasons favorable to good vegetative performance [48]. The mathematical adjustments showed linear seedling height and stem diameter growth, evidencing that the seedlings maintained linear development and expressed regularity in the face of environmental conditions. However, growth rates decreased with increased water restriction (Appendix A).

Lima et al. [49] found a 38% reduction in the growth of the aerial part of ipê-rosa seedlings when increasing the number of days between waterings during plant hardening. In addition, these authors observed effects on the metabolic pattern of seedlings subjected to water restriction, with substantial phenotypic plasticity during acclimatization to field conditions and with preferential redirection of photoassimilates to the root regions of the plant. Plant growth is generally represented by a sigmoid curve, modeled by phases with more or less accentuated productivity according to the plant’s phenophase and the environment in which it is inserted [50,51]. However, forest seedlings present little phenological distinction in nursery conditions, given the short permanence period and growth environment with restrictive characteristics [52]. Thus, there was constant growth at the same level of water replenishment, with the rate of increase being influenced by the availability of water in the substrate.

Growth assessments were carried out after the beginning of the rainy season in the region when the temperature and relative humidity were at levels that were biologically more favorable to plant development than the dry season (Figure 4). This environmental condition, associated with the protected environment of plant production, may also have favored the regular growth in height and DAC of the ipê-rosa seedlings. Excessive temperature increases and relative humidity reductions antagonistically affect the assimilation of CO_2_ used in biomass formation by plant species, as they alter the activity of enzymes that act in the photosynthetic pathways and the electron transport chain. In this case, there may be a reduction in the degree of opening of the stomata and an increase in the O_2_/CO_2_ ratio dissolved inside the *Chl*oroplasts, which in turn increases the photorespiration rates (a route that occurs due to the ability of the Rubisco enzyme—a carboxylation agent—to use oxygen as a substrate); in addition, there may be an increase in respiratory activity, with the use of assimilated organic compounds to maintain cellular biochemical integrity [53,54]. Vieira et al. [5], when evaluating the growth of ipê-rosa in its juvenile phase, grown at different levels of base saturation in the soil in a greenhouse for 90 days, indicated a height of 23.3 cm as ideal for transplanting in the field, also considering technical and financial perspectives. Under the conditions of the present study, thermal accumulations of 1344, 1056 and 979 DDAs would be required for the ipê-rosa seedlings to become suitable in height to leave the nursery conditions for 50, 75 and 100% LWRs. Notably, for seedlings with the minimum LWR (25%), around 4940 DDAs would be necessary to reach the aforementioned morphometric standard.

The numbers of leaves and leaflets showed different behaviors depending on the thermal sum, indicating the existence of accumulated thermal sums that provide maximum and minimum numbers of leaflets, respectively. Leaf tissues represent the basis of the photosynthetic process and are generally the first parts of the plant body to respond to the appearance of unfavorable conditions in the growth environment, demonstrating regulated ontogeny and morphophysiology to minimize irreversible damage to the metabolic integrity of the plant, while also seeking to maximize photoassimilative processes [21]. The ipê-rosa seedlings did not begin the growth period at 25% LWR since all treatments presented the same level of water replacement (irrigation) until the thermal accumulation of 345 DDAs (21 DATs). In this case, these plants initially captured the condition of water restriction and adjusted their physiology to a lower LWR to assume a new growth rhythm, demonstrated by the number of leaves and leaflets. For the other morphometric components of these same seedlings, it would probably be necessary to extend the analyses in time for growth adjustments to occur as long as the other environmental conditions were maintained. The gradual separation in the curves of the responses to LWR for seedling height, stem diameter at height and number of leaflets showed us the negative effect of restricted water supply on the growth of young ipê-rosa plants. Thus, we can state that juvenile ipê-rosa can tolerate short-term water stress and maintain its growth rate. However, if this unfavorable water condition is sustained, the species tends to regulate its morphological structure to the margin of survival; that is, it reduces stem growth and leaf senescence to maintain the physiological stability of the plant until water is re-established in the environment [15]. Furthermore, water plays a fundamental role in plant growth because it is associated with the activity of signaling hormones and protein groups in the cell wall, enabling the process of cell expansion. Its absence gradually makes cell elongation impossible and partially stops the tissue growth cycle [55,56].

The stagnation in the gain in leaf number of seedlings subjected to 100% LWR was not qualified by the “turnover” of the photosynthetic apparatus. According to Monteiro [21], this moment is defined by the emergence and senescence of equivalent leaves, renewing the photoassimilative capacity of forest seedlings. In this case, leaflets were more pronounced at this 100% LWR, indicating the anticipation of the leaf ontogeny of ipê-rosa cultivated at maximum water replacement.

At the end of the experiment (at 107 DATs, with an accumulation of 1655 DDAs), the influence of water restriction on the accumulation of seedling mass was evident, with a significant reduction for the 25% replacement level. Lima et al. [49] found a 50% loss in biomass input in ipê-rosa seedlings subjected to water deficiency compared to irrigated plants of the species. Water restriction reduces the carbon fixation capacity of plants by affecting multiple pathways that make up the photosynthetic chains dependent on the presence of water, such as changing the balance between cyclic and acyclic photophosphorylation, in addition to inducing the occurrence of alternative processes of stress tolerance, maintenance of internal water content, and cellular integrity and homeostasis; these effects are also transmitted to the morphological structure of the plant through the relationship with the allocation of carbon in the aerial and root parts [57,58,59]. The change in leaf area due to water availability, in turn, is a mechanism associated with reducing the leaf surface in contact with the atmospheric stratum to reduce water loss through transpiration, which is recurrent in different higher plants [60]. However, the marked loss in leaf area observed in 25% water replacement was probably more related to insufficient carboxylation than to the phenotypic expression of tolerance of ipê-rosa seedlings under water stress. This finding is reinforced by the patterns of dry mass partition between parts at LWRs (Figure 6), since regardless of the amount of water in the substrate, there was regularity in the proportions of area and root dry mass contribution after 107 DATs. Also, the ipê-rosa seedlings demonstrated a single phenotypic category for the distribution of photoassimilates among the plant organs, unaltered by the water status of the growth environment, which was based on leaf production.

Plants in the Amazon biome acclimate to drought, with adjustments observed throughout the metabolic chain to minimize permanent damage to the photosynthetic apparatus. Once the water regime is resumed, vegetative performance tends to reach the “optimal” level, indicating resilience to extreme climate events [61]. The ipê-rosa seedlings demonstrated the ability to adjust their morphological structure to the survival margin of water deficit; consequently, they expressed a gradual reduction in growth variables with the reduction in water availability in the substrate.

### 4.2. Water Requirements and Sensitivity of the Ipê-Rosa Seedlings 

The climatic seasonality during the experiment caused changes in the crop evapotranspiration (ETc) and reference (ETo) patterns; with the onset of rains, there was a greater variation in daily values. Biophysical processes of plant interaction with the growth environment regulate daily water demand. Thus, the plant’s seasonality depends on water energy balances, since plants can adjust their morphophysiology to microclimatic seasonal patterns [16,62]. A continuous increase in the water demand of the seedlings of this species was observed, accompanying the increase in the transpirational surface and morphometric variables in the seedling cycle. Although forest seedlings do not present defined phenological phases, water requirements are regulated by the plant’s growth stage, culminating in an increase in potential evapotranspiration with increased allocation to plant organs, thus requiring more and more water to maintain good plant performance and homeostasis [63].

The size of seedling growth containers (tubes, pots, plastic bags, among others) can generate different growth rates for ipê-rosa seedlings, which can interfere with potential evapotranspiration. According to Cunha et al. [64], in the volumetric range of 2.7 to 11.5 L, there are gains in height, stem diameter, and aerial and root biomass of ipê-rosa seedlings, since larger containers, associated with other components of forest seedling management, tend to improve the vegetative performance of this species by up to 76%. The crop coefficient is an index that facilitates irrigation management of agricultural and forestry crops, associating it with easily obtainable climate variables [65]. However, few studies in the silvicultural field quantify kc values as a function of the thermal requirements for seedlings of native Amazon biome species, reinforcing the present study’s importance. However, there is still a demand for research that associates these responses with other components essential to the success of production in forest nurseries, such as container size, substrate quality and level of shading, among others.

Regarding the degree of tolerance of seedlings to water conditions (Ky), although no linear behavior of Ky was observed as a function of water replacement, ipê-rosa seedlings were sensitive to water shortage. Pessoa et al. [66] characterized a rapid and progressive reduction in the physiological performance of ipê-rosa seedlings after irrigation was suppressed at 10-day intervals; however, the plants maintained regular water use efficiency. Keffer et al. [29] described the sensitivity to water restriction of young *Handroanthus serratifolius* plants under different shading levels, with Ky values lower than those obtained in the present study, ranging from 0.89 to 1.28. In a similar analysis of Ky, Bueno et al. [67] measured the plant growth factor (Gpf) of *Schizolobium parahyba*, *Cytharexylum myrianthum* and *Ceiba speciosa* at the seedling stage as a variable that estimates water sensitivity based on non-destructive growth parameters (stem height and diameter); for this purpose, they obtained a Gpf below 0.5, classifying the seedlings of these species as having low sensitivity to water restriction. In addition to the degree of incident solar radiation and water availability, other technical and ecophysiological components, such as container size and substrate quality, can regulate the biomass yield of forest seedlings and, consequently, their demands on the environment [68,69]. In general, water demand is expected to increase with the vegetative development of the seedlings [63]. Plants subjected to 25% LWR regulated dry matter production to a restricted margin from the beginning of treatment application, while those at 50% LWR possibly had sufficient supply to regulate growth for a longer cycle period. When the growth stage requires greater water availability, it is characterized by greater water deficiency in this condition, reflected in Ky.

Studies that combine irrigation regimes, growth factors, and performance indices, such as Ky for forest nurseries, are gradually being requested. However, they are still scarce in qualifying the best cultivation conditions for conserving and managing environmental and financial resources. From another perspective, Amazonian ecosystems have demonstrated differential behaviors in recent years in the face of climate change, mainly regarding the extension of dry days, resulting in changes at the level of individuals and entire plant communities, through the regulation of organisms already established to the dynamics of the environment, as well as the recruitment of taxa more related to drought conditions [70]. Despite the expressive water sensitivity observed in the present study, Ky has few literary parameters for Amazonian tree species. Thus, ipê-rosa plants expressed a certain resistance to water shortages, with vegetative performance regulated to the survival margin, verified by the inversion of Ky between 50 and 25% LWRs.

### 4.3. Gas Exchange of Ipê-Rosa Seedlings

The region where the experiment was conducted is characterized by variations in cloudiness during the dry–rainy and rainy transition periods [45], and these atmospheric dynamics influenced the micrometeorological variables of the protected production environment and, consequently, the gas exchange of the ipê-rosa seedlings throughout the day. The seasonality and diurnal variation of the growth environment affect the physiological responses of evergreen species in different production systems, with the circadian rhythm of the plant’s water status indicators responding together with the micrometeorological variables, which enhance the occurrence of maximum gas exchange values at solar noon, associated with maximum air temperature, water vapor pressure deficit and PAR radiation [71].

Marenco et al. [72] evaluated the diurnal photosynthetic fluxes of leaves of seedlings of ten Amazonian Forest species under the forest canopy in consideration of regional rainfall seasonality (dry and rainy). They observed that, regardless of the season, the net photosynthesis rate in saturating light increased until local noon, with a reduction in the afternoon. According to these authors, in the shaded condition under the canopy, the photosynthetic activity of these emerging plants is only favored when a greater amount of light enters the environment, associated with higher temperatures that allow better enzymatic dynamics, since, in both seasons (dry and rainy), water availability does not become a restrictive element in native forests.

In the ipê-rosa seedlings evaluated, stomatal conductance did not present significant differences between the levels of water replacement at solar noon. The degree of stomatal opening of Amazonian species is expected to respond directly to changes in leaf photosynthesis and transpiration rates [72]. However, the leaf anatomy of the ipê-rosa may have contributed to the control of stomatal activity since this species is classified as hypostomatic, i.e., stomata are mostly dispersed in the abaxial portion of the leaf organs, thus minimizing their responses to the influence of direct solar radiation [73]. However, for the leaves of seedlings with 100% water replacement, at solar noon, stomatal conductance showed maximum values concurrent with peaks of A and minimum values of Ci/Ca. Therefore, internal carbon reduction was assumed to be closely associated with CO_2_ consumption in photoassimilative processes. For plants with 25% LWR, the minimum Ci/Ca ratio at local noon may be associated with the stability of gs and A; in this case, CO_2_ was probably released through respiration and photorespiration processes due to the high temperature and low dPv.

Despite the antagonistic nature of respiration and photorespiration to the effective assimilation of inorganic CO_2_, given the increase in the proportion of sugars destined for metabolic regulation and, consequently, the reduction in their availability for growth [74], both routes are essential to avoid cellular collapse, enabling acclimatization and maximum physiological integrity of the plant during stress [54]. The gas exchanges observed for ipê-rosa seedlings in a greenhouse provide a central alternative based on diurnal micrometeorological dynamics: C3 plants, such as ipê-rosa, regulate photosynthesis according to the combination of extremes related to air temperature and water vapor since values below or above the “optimum” induce restrictions in four crucial processes of inorganic CO_2_ assimilation. In this case, the following can be highlighted: (i) the capacity of the enzyme ribulose 1,5-bisphosphate carboxylase/oxygenase (Rubisco) to consume ribulose-bisphosphate (RuBP) molecules; (ii) the regenerative potential of RuBP in reactions adjacent to the Calvin Cycle and thylakoid membrane; (iii) the capacity to synthesize sugars for the consumption of ATP and regeneration of inorganic phosphate; (iv) and the transport of electrons in the conversion of light into chemical energy [53]. Furthermore, in plant organisms, the thermal index of the mesophyll defines the cellular metabolic rhythm; within certain limits, the increase in leaf temperature implies the favoring of carboxylation processes, which, on the other hand, is associated with greater water loss for regulation of physiological integrity [53]. However, studies are still needed to characterize this species’ ideal environment for photosynthesis based on other micrometeorological conditions.

Another perspective to be discussed refers to the depletion of water available for the metabolism of the plants evaluated. Despite the generalized reduction in E and A in all water regimes after 12:30 h, the responses of the leaves subjected to 100% LWR remained above the other treatments throughout the afternoon period, indicating that the availability of water in the substrate was consistent with that required by the evapotranspiration demand of the seedlings. Thus, in this reference condition (100% LWR), there were no restrictions on the amount of water available, and the seedlings presented losses linked to “potential evapotranspiration”. On the other hand, the other treatments enhanced photosynthetic activity during the morning period. Gas exchange is extremely sensitive to water availability since the lack of this resource stimulates the regulation of plant physiology to prevent water loss and the occurrence of successive damage to its biomolecular, anatomical and, ultimately, morphological structure and cellular homeostasis [18].

Dombroski et al. [75], when suppressing irrigation of ipê-rosa seedlings, found a combined negative effect on A, E and gs from the third day in the restrictive environment; however, photosynthetic activity was quickly able to recover after the re-establishment of the irrigation regime. In the present study, throughout the day (hours), the lack of statistical differences for A within the extremes of water supply (25 and 100% LWRs) indicates that environmental changes in the daily flow were insufficient to affect the assimilative stability of these plants. However, under conditions of 50 and 75% LWRs, photoassimilation was favored by the water availability of the morning period; even so, the EUA remained unchanged. This species has a high capacity to use water in the photoassimilative process, modifying the morphophysiological structure and increasing its efficiency according to the restrictive impositions of the growth environment [27]. In this sense, treatments with lower water replacement were expected to demonstrate higher average WUE values in consideration of the complete compensation of daily evapotranspiration. Thus, it is assumed that, at the genotypic level, the evaluated ipê-rosa seedlings had an ideal margin for WUE, regardless of water replacement, associated with adjustment based on water depletion in the substrate at the end of the day. Different progenies tend to react differently when subjected to environmental stress, altering the phenotype of their morphophysiology according to the genetic structure previously adjusted to the native place of their growth and reproduction [19]. This effect can be even more pronounced when we deal with forest species cultivated in a seed form, subject to intraspecific genetic variation; starting from the same mother tree, there are multiple genetic groups [76].

Seedlings of Amazonian tree species show a high capacity to adjust gas exchange in leaf organs in response to environmental stress [77]. When subjected to drought, these plants demonstrate a gradual reduction in photosynthesis, accompanied by alternative mechanisms to carbon and water flows throughout the plant and the development of survival and vegetative resilience strategies. Once regularly rehydrated, they maximize the carbon assimilation process and regulate other indices close to the “optimum”, as long as the other components of the environment are not affected [77]. The leaves of ipê-rosa seedlings reacted similarly at metabolic extremes in comparing the gas exchange of seedlings with 25 and 100% LWRs. Given the diurnal micrometeorological changes, the leaves at 25% water replacement leveled their photosynthetic activity downwards, differently from what was observed for the seedlings at 100% water replacement. On the other hand, the leaf organs acclimated to 50 and 75% LWRs showed greater environmental sensitivity as mentioned above; assimilation was intensified during the morning period.

Finally, the species’ genetic set possibly showed the capacity to adjust its leaves’ photosynthetic physiology to water deficiency since the variables evaluated concerning adaptability to the growth condition, EUA and TF, remained similar in the diurnal flow, even though the amounts of water replacement were different throughout the experimental period.

### 4.4. Chl a Fluorescence of Ipê-Rosa Seedlings

The literature describes that higher plant organisms, under non-stress conditions (biotic or abiotic), present an Fv/Fm close to 0.83 [78]. This behavior can be observed in the evaluated ipê-rosa seedlings since, initially, at 28 DATs (437 DDAs), the variations in the amount of water replaced in the substrate did not alter the integrity of the photochemical apparatus of the leaves. With the evolution of the plant cycle (increase in thermal sum), it was noted that only the plants subjected to 25% LWR underwent photochemical stress. At the same time, the other treatments maintained the integrity of the PSII reaction centers. It was expected that the reduction in water replacement levels would cause a gradual reduction in Fv/Fm_predawn_ after the plants acclimated to the respective treatments [36]; however, this was not observed. Yin et al. [19] also did not observe a decrease in Fv/Fm in *Jatropha curcas* L. seedlings grown in soil at 30% of field capacity compared to those treated at 50 and 80% of field capacity; these authors attributed the differences, not classified as photochemical stress, to the origin of the genetic materials used, which, in turn, demonstrated different phenotypic expressions as an effect of the respective growth environments.

The biochemical functioning of the structure of the dimeric protein supercomplex with multiple subunits of PSII in higher plants is resistant to water stress when this is not associated with other factors that limit good vegetative performance (such as high levels of light and salinity), given the nature of photoprotective mechanisms capable of dissipating excess energy through alternative non-photochemical extinction pathways [79]. Furthermore, Elsheery and Cao [36] suggest that the reaction centers may not be closed and that the accumulation of irreparable damage to the protein complexes associated with the conversion of solar energy into chemical energy may be avoided, presumably through the de-epoxidation of the pigments of the xanthophyll cycle. Therefore, given that photochemical stress was characterized only in the 25% LWR, biochemical and physiological acclimatization of the photochemical apparatus of the plants occurred in water replacement of 50% of that required by daily evapotranspiration or higher, which is sufficient to maintain the functioning of the protein complexes that make up the photochemical chains (PSI, PSII, Cytochrome B6F and ATP synthase).

In addition, the protective potential of plant tissues against damage to the photochemical apparatus, preceding the effective closure of the reaction centers responsible for regulating the maximum support of photochemical energy to ideal levels [80], may have triggered the late impact of water restriction (25% LWR) on Fm_predawn_, which was observed only at 90 DATs (1400 DDAs).

The diurnal flow of Fv/Fm, with acclimation to darkness for 30 min, exposed at 28 DATs (463 DDAs), showed little variation as a function of water availability, probably affected by the genetic variability inherent in seedlings from seed propagation, before water replacement. Despite the influence of water treatments on subsequent evaluations, as well as the distinct variations in the growth environment, the daily dynamics of Fv/Fm for ipê-rosa were related to micrometeorological patterns, outlining curves with a downward concavity, whose minimum points occurred with the maximum daily photosynthetically active radiation, between 11:00 and 13:00 h. Marenco et al. [74] observed a similar pattern for Fv/Fm in leaves in the upper canopy of the species *Coussapoa orthoneura* and *Protium opacum*, with a reduction from 0.80 to 0.70 at midday and subsequent recovery in the late afternoon. The recovery of the Fv/Fm variable values after the midday depression in Amazonian plants is associated with the adaptive potential of these organisms in the face of diurnal climate extremes, characterizing the existence of dynamic photoinhibition [54]. Specifically, there is an increase in energy dissipation in the form of heat and the occurrence of reversible damage to the photochemical apparatus during the stress period; after the reduction in radiation levels and air temperature, there is a gradual temporary closure of the xanthophyll cycle and reactivation of damaged PSII proteins [81]. Analogous to gas exchange, due to the influence of the micrometeorological dynamics during the day inherent to the period in which the experiment was conducted, there was less ordering in the values of Φ_PSII_, ETR, Fv′/Fm′, NPQ and qL in consideration of Fv/Fm throughout the diurnal evolution.

The variables that make *Chl a* fluorescence in the light-adapted state provide efficiency in coordinating the operation of the photochemical apparatus with protein complexes acclimated to the current growth environment without the “turnover” of the chains and cycles of the photosynthetic machinery or complete oxidation of the primary electron acceptor—QA—and are regulated by momentary fluctuations in the microclimate in a more pronounced way than Fv/Fm, which in turn is determined after 30 min of dark adaptation [39]. However, at 90 DATs (1400 DDAs), the Φ_PSII_ and ETR of the ipê-rosa seedlings with water availability of 50% water replacement or higher demonstrated an increasing linear trend since the levels of air temperature, dPv and PAR being lower than those of the other evaluation dates gradually made the environment more favorable to the photochemical stage of photosynthesis. The seedlings under 25% LWR showed a linear decreasing trend in the variables mentioned on the same date, with results superior to those obtained in previous evaluations in this water condition. These plants obtained better photochemical performance in the milder environment; even so, the increase in air temperature, dPv and PAR after 11:00 h was also associated with the depletion of water available in the substrate, causing a reduction in Φ_PSII_ and ETR.

Plant organisms tend to present different phenotypes depending on the interaction between air temperature, available water, dPv and solar irradiance, altering their morphophysiological structure to survive stress, accompanied by photoprotective mechanisms and the partitioning of the flow of absorbed electrons into assimilative and non-assimilative processes of chemical energy [21]. Furthermore, the functioning of the photosynthetic apparatus is regulated under the optimum limits of environmental factors, air temperature and irradiance. Quantities below these horizons demonstrate insufficiency for the perpetuation of photochemical activity. On the other hand, exceeding them triggers the occurrence of photoinhibition, in principle, in a non-lethal way to the tissues under the condition [82]. Reinforcing this discussion, it is clear that enzymatic activity is regulated by the balance between the components necessary for the occurrence of biochemical reactions, that is, substrate availability and other factors and cofactors that characterize the good performance of a given enzyme [57]. Within the electron transport chain, PSII works in three stages: capturing light energy, converting it into chemical energy and oxidizing the water molecule, triggering the formation of NADPH molecules and ATP synthesis [83].

Thus, the relationship between water availability and irradiance levels, associated with other environmental patterns, qualifies the performance and balance between the energy dissipation pathways of PSII [17]. Concomitant with the micrometeorological variations at 90 DATs, the leaves of the ipê-rosa seedlings maintained at 100% water replacement had the potential amount of water based on the daily evapotranspiration demand. Thus, the passage of the flow of photosynthetic photons through the electron transport chain, continuously generating NADPH and ATP, controlled the demand for the photochemical activity of PSII to the maximum. When already acclimated to the level of 25% LWR, at 66 and 90 DATs, the ipê-rosa seedlings exhibited the lowest percentage variability for NPQ compared to the other treatments.

The physiological expression observed in the leaves of these seedlings provides two lines of analysis that suggest substantial plasticity in the photosynthetic apparatus of ipê-rosa plants in the juvenile phase: (i) seedlings under water stress maintained greater stability in their photochemical activity due to the lower need to change their physiology for survival within micrometeorological extremes; (ii) on the contrary, individuals grown at regular to optimal levels of water replacement responded to environmental limitations to a greater extent, demonstrating tolerance and high photochemical recovery capacity [61,84]. In the first analysis, the fraction of open reaction centers of PSII (qL), which estimates the capacity of the photochemical apparatus to receive electrons considering the complete oxidation state of QA, classifies the degree of impairment of this protein complex [39].

Phytochrome interconversion, regulated by red light wavelengths, directly influences the photochemical activity of PSII. In contrast, the 600 to 700 nm (red) range, which is prominent at the beginning and end of the day, induces activation of the photosynthetic machinery. In comparison, the far-red range (from 730 nm onwards) induces activation of the photosynthetic machinery; in this case, in the final moments of sunlight, PSII “turnover” occurs until the beginning of the night, characterized by a decrease in the action of the electron acceptor and donor fractions of PSII [85]. However, despite the “closure” of the reaction centers, the wavelengths that make up the far-red are essential for maintaining the integrity of PSII; exposure of leaf tissues to these wavelengths drives the regeneration of the proteins in the complex, with subsequent improvement in photochemical capacity [86]. Depending on the times and dates of subsequent assessments, qL values increased until 1:00 p.m. Ouzounis et al. [87] observed opposite behavior for qL in young *Lactuca sativa* plants, with a reduction in qL due to the gain in radiation incident on the leaf surface. This confirms the ability of young ipê-rosa plants to regulate their photochemical activity in the different water replacement treatments according to the characteristics of the growth environment. The perpetuation of photoprotective mechanisms ensured the regulation of the degree of opening of the photoreceptor channels based on the integrity of PSII as a whole, controlling the fractionation between photoassimilative or non-photoassimilative dissipation routes [21,36].

Under general environmental conditions, the variables that make up the *Chl a* fluorescence in the light-adapted state of ipê-rosa seedlings responded partially to microclimate variations during the diurnal periods evaluated, depending on water replacement. For the Cerrado tree species *Miconia fallax*, *Didymopanax macrocarpum*, *Qualea grandiflora*, *Ouratea hexasperma* and *Roupala montana*, Franco and Lüttge [88] described a direct relationship between photochemical performance and photosynthetic photon flux density, high ΦPSII in the early morning, decreasing with increasing irradiance until the maximum depression, prolonged between 11:00 and 15:00 h, with values of 0.09 to 0.26, and upcoming regeneration at the end of the afternoon period. On the other hand, Bacarin et al. [89] demonstrated greater stability for Φ_PSII_ and Fv′/Fm′ in *Gallesia integrifolia* seedlings in the face of diurnal variations in the environment, with averages showing a small increase between 7:30 and 11:30 a.m., a sharp reduction during local noon (time of maximum air temperature and irradiance) and a successive increase until the end of the daylight hours.

The transition of the environment (end of the dry season to the beginning of the rainy season) provided distinct responses for the *Chl a* fluorescence variables in the light-adapted state of ipê-rosa seedlings at 464, 1046 and 1400 DDAs. This also indicates the adaptive degree of the species in consideration of the combination of air temperature, dPv and PAR radiation associated with water availability in consideration of photosynthetic integrity and performance. The physiological patterns of higher plants respond to the combination of environmental factors once adjusted to a certain stress circumstance, regulating their metabolic activity to the margin of survival, being susceptible to the occurrence of processes antagonistic to cellular homeostasis due to the prolongation of unfavorable conditions [90].

Ipê-rosa is a species dispersed across multiple Brazilian biomes, and as such, it demonstrates the ability to adjust its phenotypic expression according to local environmental variations [2]. In its juvenile phase, it requires a substantial period in a given environment to effectively show effects on its photochemical potential and activity [91]. Given the results, only the submission of the ipê-rosa seedlings to 25% water replacement (compared to the daily potential evapotranspiration) resulted in damage to the PSII reaction centers, which, although not very high, characterized the existence of photochemical stress and chronic photoinhibition. Therefore, there were metabolic adjustments, as well as gene expression that conferred a certain degree of tolerance to water restriction on the part of these seedlings in the face of changes in the microclimate, with the continuous adjustment between the dissipation routes of solar energy intercepted by the leaf surface being combined with the balance with photoprotective chains to regulate the metabolic rhythm for survival.

Phylogenetic characteristics are essential in discerning the ideal environment for photosynthetic integrity in Amazonian species, not only defining the ideal environmental levels to guarantee good vegetative performance but also influencing the ability to resist environmental adversities, mainly regarding the combination of irradiance, air temperature and water availability [92,93]. Multiple studies attest to the effectiveness of using *Chl a* fluorescence as a tool for monitoring water stress in plants, allowing the identification of management practices that lead to good silvicultural and agricultural results [94,95]. In this sense, the water supply to ipê-rosa seedlings grown in a protected environment with replacements greater than 50% of the daily potential evapotranspiration was sufficient to keep them out of the state of water stress.

## 5. Conclusions

Ipê-rosa seedlings regulate the physiology of growth, gas exchange and *Chl a* fluorescence according to the amount of water available. Only a 25% level of water replacement (LWR) in the substrate allows the seedlings to survive.

The growth in height and diameter at the collar decreases according to the water restriction, with 90.9 and 84.7% decreases in the increase in these variables in ipê-rosa seedlings with 25% LWR compared to those with 100% LWR. The biomass accumulation of seedlings responds with a reduction of 96.9% compared to the same water levels.

Ipê-rosa, in a greenhouse, with a thermal accumulation of 1655 degree days, presents an accumulated crop evapotranspiration of 593.20 mm and a crop coefficient (kc) that increases over time, varying from 0.59 to 2.86 during the seedling cycle, with an average kc of 1.75. The water sensitivity of young ipê-rosa plants is inverted, with the highest Ky occurring at 50% LWR.

The gas exchange variables (net photosynthesis, transpiration and stomatal conductance) decrease throughout the day in 50% and 75% LWR conditions. However, during the diurnal flow, the lowest values of these variables occur in the treatment of 25% LWR, and the highest at 100% LWR.

A water supply of 50% of daily evapotranspiration keeps the leaves of ipê-rosa seedlings out of photochemical stress. In the long term, only 25% LWR of this quantity regulates them for survival, with the occurrence of non-photoassimilative energy dissipation pathways.

Regardless of water availability, the maximum quantum yield of PSII in ipê-rosa leaves responds inversely to diurnal variations in the microclimate, reaching minimum values at times of maximum daily photosynthetically active radiation.

The variables that make up the *Chl a* fluorescence of ipê-rosa seedlings in the light-adapted state (Φ_PSII_, ETR, Fv′/Fm′, NPQ and qL) demonstrate different physiological responses to the combination of microclimatic factors and available water.

## Figures and Tables

**Figure 1 plants-13-02850-f001:**
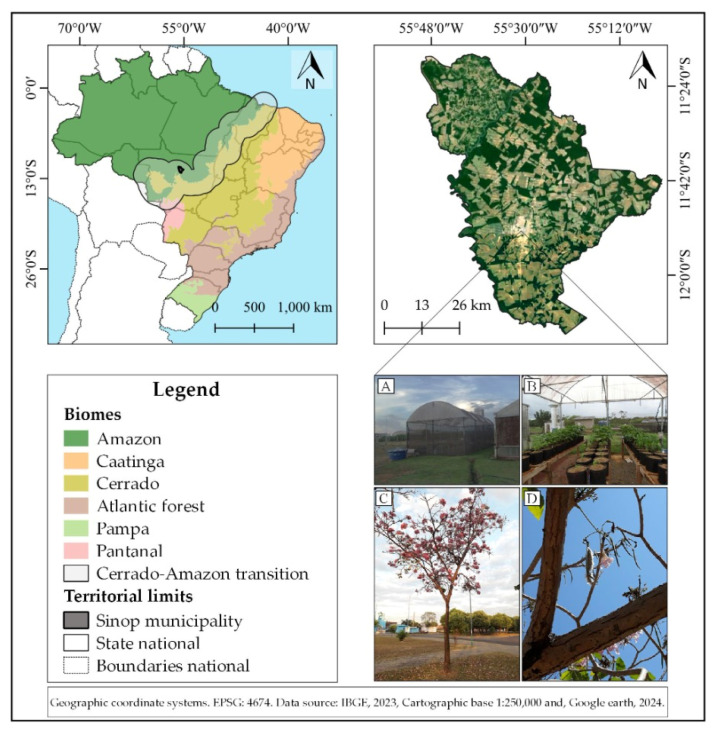
Location map of the collection region of *Handroanthus impetiginosus* seeds, Sinop, Mato Grosso state, Brazil. (**A**) plastic greenhouse for producing forestry agricultural seedlings; (**B**) view of seedlings in pots; (**C**) adult ipê-rosa tree in bloom; (**D**) ipê-rosa pods and seeds.

**Figure 2 plants-13-02850-f002:**
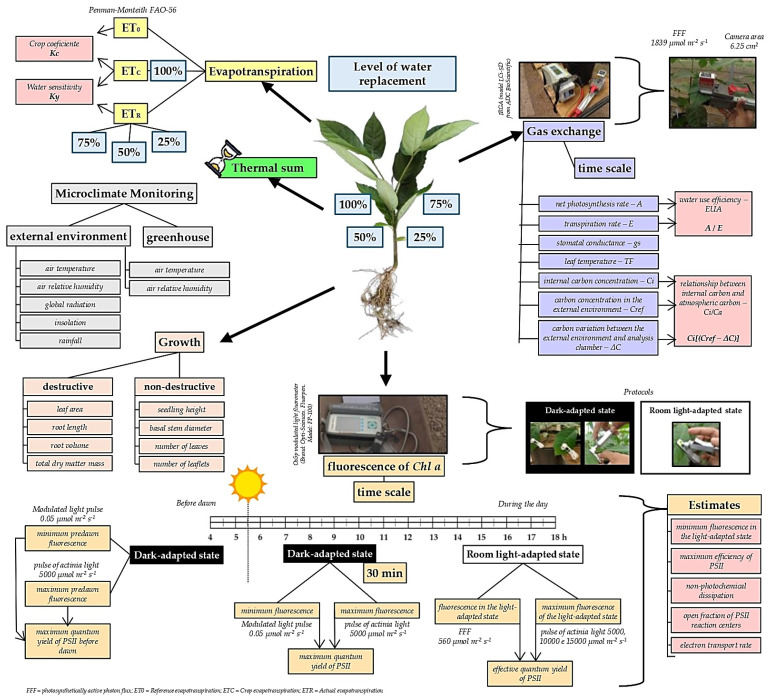
Flowchart of the experimental steps used in the development of the research. All symbols are explained throughout the sub-items of the methodology.

**Figure 3 plants-13-02850-f003:**
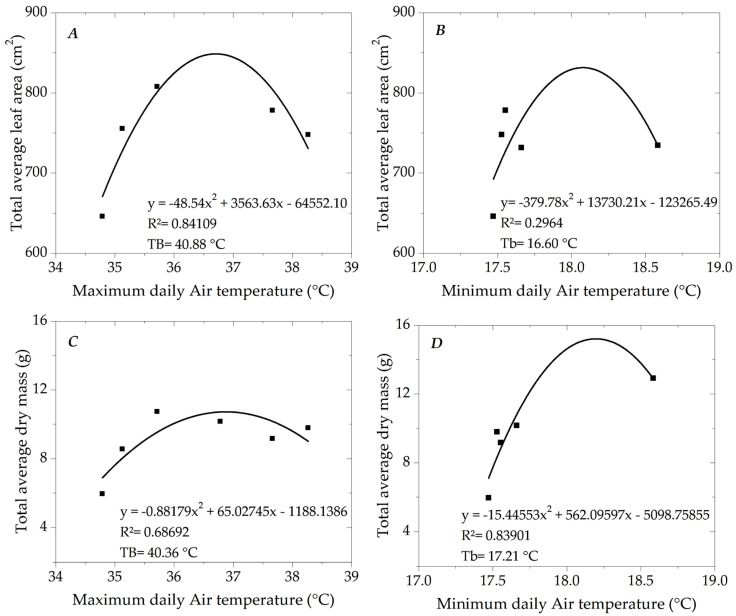
Correlation between maximum and minimum air temperatures and the total average leaf area (**A** and **B**, respectively) and total average dry mass (**C** and **D**, respectively) per plant to determine upper basal temperature (TB) and lower basal temperature (Tb), in Sinop, Mato Grosso state, Brazil. Prepared from the database assembled by Monteiro [40].

**Figure 4 plants-13-02850-f004:**
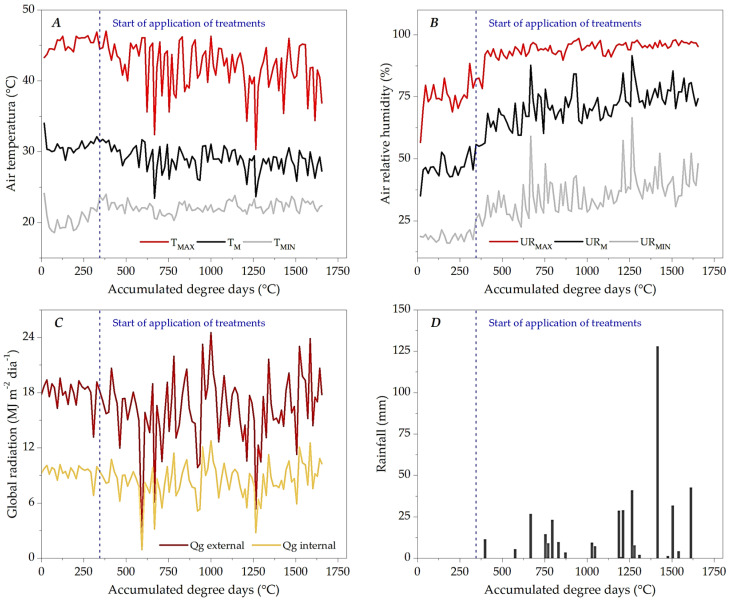
Daily variations in maximum (T_MAX_), average (T_M_) and minimum (T_MIN_) air temperature (**A**) and maximum (RH_MAX_), average (RH_M_) and minimum (RH_MIN_) relative air humidity (**B**) outside the greenhouse; global radiation outside and inside the greenhouse (**C**); and rainfall (**D**) as a function of the accumulated degree days of the ipê-rosa seedlings, between 28 August 2019 and 13 December 2019, in Sinop, Mato Grosso state, Brazil.

**Figure 5 plants-13-02850-f005:**
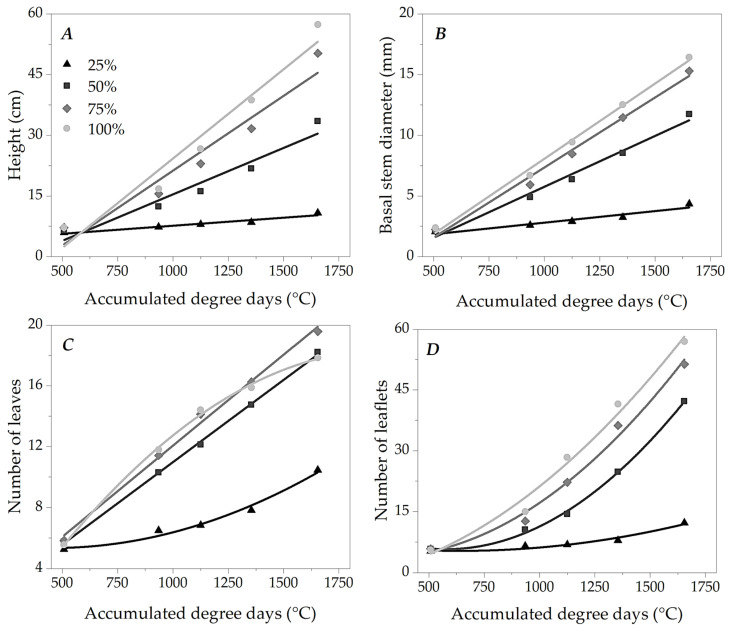
Adjusted regression curves for the non-destructive growth variables height (**A**), basal stem diameter (**B**), number of leaves (**C**) and number of leaflets (**D**) of ipê-rosa seedlings at 25, 50, 75 and 100% levels of water replacement (LWRs), as a function of the accumulated degree days, in a greenhouse.

**Figure 6 plants-13-02850-f006:**
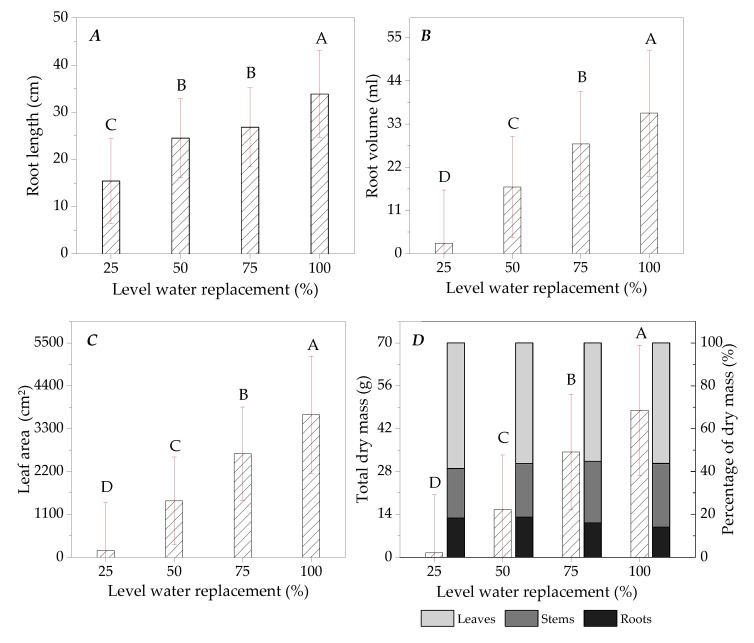
Mean values of root length (**A**), root volume (**B**), leaf area (**C**), total dry mass and percentages of dry mass partition of leaves, stems and roots (**D**) of ipê-rosa seedlings at 25, 50, 75 and 100% levels of water replacement (LWRs), 107 days after transplanting, in a greenhouse. Means followed by the same capital letter between water replacement levels do not differ by Tukey’s test at the 5% probability level.

**Figure 7 plants-13-02850-f007:**
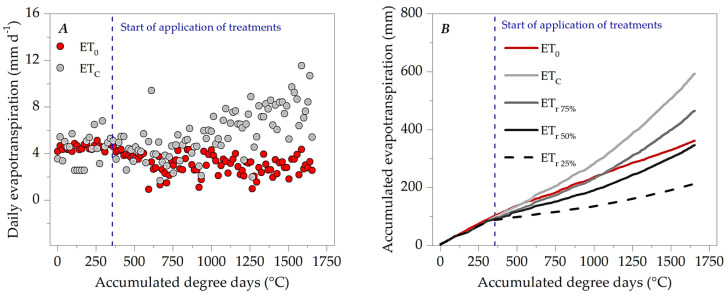
Daily evapotranspiration (**A**) and accumulated evapotranspiration (**B**) for the crop (ET_C_: ipê-rosa seedlings, in a greenhouse and potential; ET_0_: Penman–Monteith FAO-56 method [34]) as a function of the accumulated degree days of the seedlings, between 28 August 2019 and 13 December 2019, in Sinop, Mato Grosso state, Brazil. Real accumulated crop evapotranspiration (ETr) under 25, 50 and 75% levels of water replacement (LWRs) is represented by ETr_25%_, ETr_50%_ and ETr_75%_, respectively.

**Figure 8 plants-13-02850-f008:**
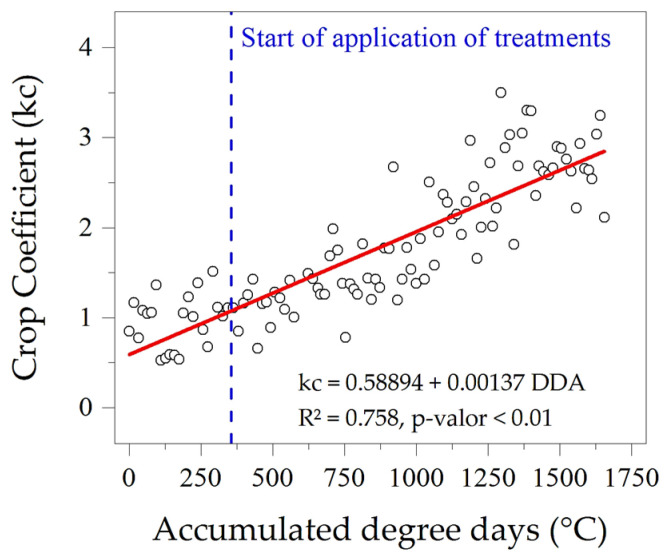
Crop coefficient as a function of accumulated degree days of the ipê-rosa seedlings, in a greenhouse (*p*-value < 0.01 indicates the significance of the adjustment at the 1% probability level; the red line represents the line of the fitted linear equation).

**Figure 9 plants-13-02850-f009:**
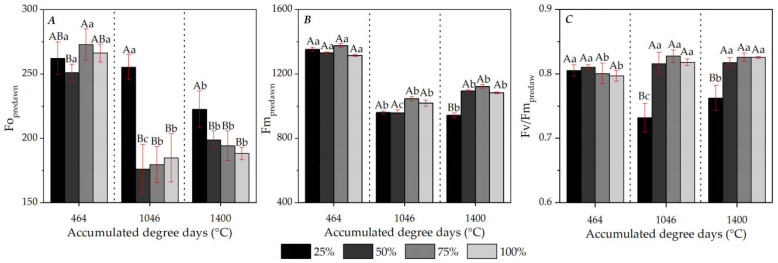
Mean values and standard deviation of minimum fluorescence (Fo_predawn_) (**A**), maximum fluorescence (Fm_predawn_) (**B**) and maximum quantum yield of PSII (Fv/Fm_predawn_) before dawn (**C**), in ipê-rosa seedlings as a function of water replacement levels, at 28, 66 and 90 DATs (464, 1046 and 1400 DDAs). The red lines represent the standard deviation values. Means followed by the same capital letter between levels of water replacement (LWRs) and lowercase letter between DDAs do not differ by Tukey’s test at the 5% probability level.

**Table 1 plants-13-02850-t001:** Water sensitivity of ipê-rosa seedlings at 75, 50 and 25% levels of water replacement (LWRs), after 107 days after transplanting, in a greenhouse, Sinop, Mato Grosso state, Brazil.

LWR	Etr ^1^	Etm ^2^	(1 − ETr/ETp)	Yr ^3^	Ym ^4^	(1 − Yr/Yp)	Ky ^5^
	(mm)			(g m^−2^)			
75%	465.41	593.2	0.22	802.47	1121.32	0.28	1.32
50%	346.15	593.2	0.42	364.27	1121.32	0.68	1.62
25%	212.99	593.2	0.64	34.4	1121.32	0.97	1.51

^1^ Etr—real accumulated evapotranspiration; ^2^ ETm—maximum accumulated evapotranspiration in the treatment with 100% LWR, representing ET_C_; ^3^ Yr—actual dry mass yield; ^4^ Ym—maximum dry mass yield in the treatment with 100% level of water replacement; ^5^ Ky—water sensitivity coefficient.

**Table 2 plants-13-02850-t002:** Net photosynthesis rate, transpiration rate, water use efficiency, stomatal conductance, internal carbon to atmospheric carbon ratio and leaf temperature at 09:30, 12:30, 13:30 and 16:00 h in leaves of ipê-rosa seedlings at 100, 75, 50 and 25% levels of water replacement, 107 days after transplanting, the thermal sum of 1557.9 DDAs, in a greenhouse.

Hour	Level of Water Replacement			
	25%	50%	75%	100%
**Net photosynthesis rate—µmol m^−2^ s^−1^**				
09:30	6.11 Ab	12.54 Aa	11.12 Aa	10.1 Aa
12:30	5.36 Ab	6.92 Bb	10.30 Aa	10.41 Aa
13:30	4.92 Aa	7.15 Ba	6.75 Ba	8.02 Aa
16:00	3.50 Ab	6.28 Bab	6.56 Bab	8.05 Aa
**Transpiration rate—mol m^−2^ s^−1^**				
09:30	2.95 Ab	5.47 Aa	4.88 Aa	5.03 Aa
12:30	2.65 ABc	3.93 BAcb	4.92 Aab	5.81 Aa
13:30	2.43 ABb	3.48 BCab	3.28 Bab	4.51 ABa
16:00	1.26 Bb	2.22 Cab	2.23 Bab	3.09 Ba
**Water use efficiency—µmol CO_2_ mol^−1^ H_2_** **O**				
09:30	2.21 Ba	2.29 Ba	2.29 Ba	1.94 Ba
12:30	2.26 ABa	1.82 Ba	2.16 Ba	1.81 Ba
13:30	1.97 Ba	2.12 Ba	2.06 Ba	1.78 Ba
16:00	2.75 Aa	2.83 Aa	3.15 Aa	2.77 Aa
**Stomatal conductance—mol m^−2^ s^−1^**				
09:30	0.10 Ab	0.23 Aa	0.19 Aa	0.21 Aa
12:30	0.07 Ab	0.10 Bb	0.14 ABab	0.19 Aa
13:30	0.06 Ab	0.09 Bab	0.09 Bab	0.16 Aa
16:00	0.06 Ab	0.13 Bb	0.13 ABb	0.24 Aa
**Internal carbon to atmospheric carbon ratio**				
09:30	0.64 Bb	0.71 Aab	0.68 ABab	0.72 ABa
12:30	0.50 Cc	0.64 Bab	0.57 Cb	0.66 Ba
13:30	0.62 Bb	0.59 Bb	0.63 BCb	0.71 ABa
16:00	0.69 Aa	0.67 Aa	0.76 Aa	0.78 Aa
**Leaf temperature—°C**				
09:30	40.37 Ba	40.03 Ba	40.40 Ba	40.81 Aa
12:30	41.94 Aa	42.02 Aa	41.97 Aa	41.90 Aa
13:30	39.53 Ba	40.41 Ba	39.47 Ba	39.43 Ba
16:00	33.63 Ca	33.47 Ca	33.83 Ca	34.19 Ca

Means followed by the same capital letter within columns and lowercase letter within rows do not differ by Tukey’s test at the 5% probability level.

**Table 3 plants-13-02850-t003:** Pearson correlation (r) between gas exchange variables net photosynthesis rate (A), transpiration rate (E), water use efficiency (EUA), stomatal conductance (gs), relationship between internal carbon and atmospheric carbon (Ci/Ca) and leaf temperature (TF) and the microclimate factors mean air temperature (T_M_) and water vapor pressure deficit of the air (dPv) in leaves of ipê-rosa seedlings at 100, 75, 50 and 25% levels of water replacement (LWRs), 107 days after transplanting, the thermal sum of 1556 DDAs, in a greenhouse.

**100% Level of Water Replacement**								
	T_M_	dPv	A	E	EUA	gs	Ci/Ca	Tf
T_M_	1.00	1.00 **	0.19 ^NS^	0.37 **	−0.52 **	−0.08 ^NS^	−0.51 **	0.78 **
dPv		1.00	0.19 ^NS^	0.37 **	−0.52 **	−0.06 ^NS^	−0.50 **	0.77 **
A			1.00	0.93 **	0.07 ^NS^	0.86 **	0.13 ^NS^	0.28 *
E				1.00	−0.28 *	0.75 **	0.02 ^NS^	0.56 **
EUA					1.00	0.15 ^NS^	0.24 ^NS^	−0.78 **
gs						1.00	0.55 **	−0.03 ^NS^
Ci/Ca							1.00	−0.61 **
Tf								1.00
**75% Level of Water Replacement**								
	T_M_	dPv	A	E	EUA	gs	Ci/Ca	Tf
T_M_	1.00	1.00 **	0.51 **	0.61 **	−0.39 **	0.24 ^NS^	−0.45 **	0.76 **
dPv		1.00	0.52 **	0.61 **	−0.37 **	0.25 ^NS^	−0.45 **	0.75 **
A			1.00	0.85 **	0.00 ^NS^	0.80 **	−0.22 ^NS^	0.41 **
E				1.00	−0.48 **	0.73 **	−0.19 ^NS^	0.70 **
EUA					1.00	−0.09 ^NS^	0.05 ^NS^	−0.68 **
gs						1.00	0.28 *	0.09 ^NS^
Ci/Ca							1.00	−0.58 **
Tf								1.00
**50% Level of Water Replacement**								
	T_M_	dPv	A	E	EUA	gs	Ci/Ca	Tf
T_M_	1.00	1.00 **	0.57 **	0.67 **	−0.65 **	0.43 **	−0.15 ^NS^	0.75 **
dPv		1.00	0.55 **	0.66 **	−0.63 **	0.41 **	−0.16 ^NS^	0.74 **
A			1.00	0.83 **	−0.24 ^NS^	0.86 **	0.13 ^NS^	0.41 **
E				1.00	−0.68 **	0.77 **	0.11 ^NS^	0.71 **
EUA					1.00	−0.22 ^NS^	0.13 ^NS^	−0.87 **
gs						1.00	0.57 **	0.17 ^NS^
Ci/Ca							1.00	−0.49 **
Tf								1.00
**25% Level of Water Replacement**								
	T_M_	dPv	A	E	EUA	gs	Ci/Ca	Tf
T_M_	1.00	1.00 **	0.13 ^NS^	0.28 *	−0.37 **	0.14 ^NS^	−0.29 *	0.74 **
dPv		1.00	0.13 ^NS^	0.26 *	−0.36 **	0.13 ^NS^	−0.30 *	0.73 **
A			1.00	0.94 **	0.11 ^NS^	0.95 **	−0.01 ^NS^	0.26 *
E				1.00	−0.21 ^NS^	0.94 **	0.10 ^NS^	0.45 **
EUA					1.00	−0.05 ^NS^	−0.42 **	−0.57 **
gs						1.00	0.23 ^NS^	0.19 ^NS^
Ci/Ca							1.00	−0.38 **
Tf								1.00

Pearson correlation coefficients (r) were submitted to the *t*-test; where cells in green and red colors correspond to **, significant at the 1% probability level, and *, significant at the 5% probability level, respectively, and NS indicates a non-significant coefficient.

**Table 4 plants-13-02850-t004:** Diurnal variations in the average maximum quantum yield of PSII (Fv/Fm), in the fluorescence of *Chl a* in the light-adapted state and in the average fraction of open reaction centers of PSII (qL) of ipê-rosa seedlings at the 25, 50, 75 and 100% levels of water replacement at 28, 66 and 90 days after transplanting (DATs), equivalent to the thermal sums of 464, 1046 and 1400 accumulated degree days (DDAs), respectively, in a greenhouse.

Solar Time	28 DATs (464 DDAs)				66 DATs (1046 DDAs)				90 DATs (1400 DDAs)			
	25%	50%	75%	100%	25%	50%	75%	100%	25%	50%	75%	100%
	Maximum quantum yield of PSII (Fv/Fm)											
07:00	0.799	0.797	0.800	0.800	0.722	0.799	0.809	0.810	0.761	0.819	0.825	0.828
09:00	0.759	0.763	0.753	0.755	0.663	0.765	0.767	0.770	0.731	0.800	0.804	0.813
11:00	0.735	0.742	0.736	0.736	0.634	0.684	0.729	0.748	0.734	0.805	0.810	0.819
13:00	0.723	0.740	0.740	0.734	0.658	0.749	0.765	0.781	0.730	0.805	0.806	0.815
15:00	0.757	0.763	0.762	0.749	-	-	-	-	-	-	-	-
17:00	0.799	0.799	0.792	0.789	-	-	-	-	-	-	-	-
	**Effective quantum yield of PSII (** **Φ_PSII_)**											
07:00	0.378	0.423	0.423	0.417	0.188	0.444	0.503	0.405	0.298	0.357	0.366	0.356
09:00	0.281	0.352	0.328	0.284	0.159	0.273	0.351	0.385	0.281	0.426	0.453	0.497
11:00	0.327	0.335	0.335	0.364	0.106	0.294	0.366	0.385	0.288	0.438	0.410	0.396
13:00	0.338	0.358	0.366	0.375	0.155	0.410	0.430	0.513	0.278	0.468	0.480	0.506
15:00	0.383	0.352	0.382	0.349	-	-	-	-	-	-	-	-
17:00	0.235	0.228	0.211	0.185	-	-	-	-	-	-	-	-
	**Electron transport rate (µmol m^−2^ s^−1^)**											
07:00	89.27	100.01	99.84	98.40	44.32	104.86	118.90	95.57	70.48	84.28	86.45	84.14
09:00	66.50	83.10	77.47	67.09	37.67	64.45	83.034	90.91	66.43	100.69	107.10	117.42
11:00	77.36	79.13	79.24	85.90	25.12	69.46	86.499	91.00	68.13	103.48	96.76	93.59
13:00	79.90	84.51	86.42	88.59	36.56	96.97	101.67	121.21	65.69	110.58	113.49	119.46
15:00	90.47	83.05	90.19	82.37	-	-	-	-	-	-	-	-
17:00	55.48	53.85	49.92	43.64	-	-	-	-	-	-	-	-
	**Maximum efficiency of PSII (Fv′/Fm′)**											
07:00	0.511	0.496	0.501	0.545	0.422	0.644	0.719	0.623	0.575	0.637	0.733	0.674
09:00	0.484	0.509	0.479	0.469	0.341	0.465	0.508	0.529	0.475	0.599	0.685	0.678
11:00	0.479	0.492	0.495	0.524	0.296	0.428	0.479	0.497	0.497	0.697	0.672	0.635
13:00	0.463	0.475	0.490	0.528	0.339	0.526	0.561	0.611	0.466	0.656	0.660	0.702
15:00	0.525	0.497	0.527	0.505	-	-	-	-	-	-	-	-
17:00	0.584	0.606	0.581	0.605	-	-	-	-	-	-	-	-
	**Non-photochemical dissipation (NPQ)**											
07:00	3.242	3.281	3.172	2.383	2.637	1.200	0.67431	1.638	1.432	1.611	0.742	1.356
09:00	2.379	2.155	2.366	2.523	2.939	2.774	2.268	2.070	2.109	1.698	0.967	1.086
11:00	2.069	2.015	1.897	1.646	3.390	2.050	2.005	2.046	1.873	0.859	1.123	1.681
13:00	2.116	2.208	2.017	1.519	2.892	1.932	1.601	1.346	2.274	1.188	1.206	0.874
15:00	1.903	2.285	1.965	1.998	-	-	-	-	-	-	-	-
17:00	1.876	1.642	1.807	1.494	-	-	-	-	-	-	-	-
	**Fraction of open reaction centers of the PSII (qL)**											
07:00	0.600	0.764	0.757	0.608	0.317	0.444	0.401	0.416	0.326	0.324	0.22	0.280
09:00	0.418	0.525	0.533	0.449	0.367	0.439	0.530	0.558	0.425	0.496	0.378	0.474
11:00	0.535	0.52	0.518	0.515	0.296	0.551	0.630	0.636	0.410	0.346	0.342	0.377
13:00	0.604	0.618	0.597	0.542	0.356	0.629	0.597	0.670	0.440	0.460	0.471	0.438
15:00	0.566	0.552	0.554	0.544	-	-	-	-	-	-	-	-
17:00	0.239	0.232	0.233	0.167	-	-	-	-	-	-	-	-

**Table 5 plants-13-02850-t005:** Pearson correlation (r) between *Chl a* fluorescence variables maximum PSII quantum yield (Fv/Fm), effective PSII quantum yield (Φ_PSII_), non-photochemical dissipation (NPQ), electron transport rate (ETR), maximum PSII efficiency (Fv′/Fm′) and fraction of PSII open reaction centers (qL) and microclimate factors mean air temperature (T_M_) and air water vapor pressure deficit (dPv) in leaves of ipê-rosa seedlings at levels of water replacement (LWRs) of 100, 75, 50 and 25%. Grouping of readings at 66 and 90 days after transplanting.

**100% Level of Water Replacement**								
	T_M_	DPV	Fv/Fm	Φ_PSII_	NPQ	ETR	Fv′/Fm′	qL
T_M_	1.00	0.99 **	−0.87 **	−0.04 ^NS^	0.46 **	−0.04 ^NS^	−0.72 **	0.82 **
DPV		1.00	−0.91 **	−0.10 ^NS^	0.45 **	−0.10 ^NS^	−0.76 **	0.83 **
Fv/Fm			1.00	0.17 ^NS^	−0.50 **	0.17 ^NS^	0.80 **	−0.84 **
Φ_PSII_				1.00	−0.68 **	1.00 **	0.55 **	0.18 ^NS^
NPQ					1.00	−0.68 **	−0.91 **	0.48 **
ETR						1.00	0.55 **	0.18 ^NS^
Fv′/Fm′							1.00	−0.71 **
qL								1.00
**75% Level of Water Replacement**								
	T_M_	DPV	Fv/Fm	Φ_PSII_	NPQ	ETR	Fv′/Fm′	qL
T_M_	1.00	0.99 **	−0.82 **	−0.12 ^NS^	0.57 **	−0.12 ^NS^	−0.74 **	0.84 **
DPV		1.00	−0.85 **	−0.17 ^NS^	0.59 **	−0.17 ^NS^	−0.77 **	0.84 **
Fv/Fm			1.00	0.27 ^NS^	−0.56 **	0.27 ^NS^	0.82 **	−0.81 **
Φ_PSII_				1.00	−0.70 **	1.00 **	0.60 **	0.08 ^NS^
NPQ					1.00	−0.70 **	−0.93 **	0.54 **
ETR						1.00	0.60 **	0.08 ^NS^
Fv′/Fm′							1.00	−0.73 **
qL								1.00
**50% Level of Water Replacement**								
	T_M_	DPV	Fv/Fm	Φ_PSII_	NPQ	ETR	Fv′/Fm′	qL
T_M_	1.00	0.98 **	−0.79 **	−0.36 *	0.42 **	−0.36 *	−0.70 **	0.62 **
DPV		1.00	−0.83 **	−0.39 *	0.42 **	−0.39 *	−0.72 **	0.62 **
Fv/Fm			1.00	0.50 **	−0.38 *	0.50 **	0.79 **	−0.57 **
Φ_PSII_				1.00	−0.76 **	1.00 **	0.78 **	0.07 ^NS^
NPQ					1.00	−0.76 **	−0.86 **	0.37 *
ETR						1.00	0.78 **	0.07 ^NS^
Fv′/Fm′							1.00	−0.55 **
qL								1.00
**25% Level of Water Replacement**								
	T_M_	DPV	Fv/Fm	Φ_PSII_	NPQ	ETR	Fv′/Fm′	qL
T_M_	1.00	0.99 **	−0.72 **	−0.77 **	0.58 **	−0.77 **	−0.80 **	−0.20 ^NS^
DPV		1.00	−0.73 **	−0.78 **	0.57 **	−0.78 **	−0.79 **	−0.24 ^NS^
Fv/Fm			1.00	0.79 **	−0.35 *	0.79 **	0.79 **	0.34 **
Φ_PSII_				1.00	−0.65 **	1.00 **	0.88 **	0.51 **
NPQ					1.0	−0.65 **	−0.84 **	0.20 ^NS^
ETR						1.00	0.88 **	0.51 **
Fv′/Fm′							1.00	0.06 ^NS^
qL								1.00

Pearson correlation coefficients (r) submitted to the *t*-test, where cells in green and red colors correspond to **, significant at the 1% probability level, and *, significant at the 5% probability level, respectively, and NS indicates a non-significant coefficient.

## Data Availability

Study data can be obtained upon request from the corresponding author or the first author via e-mail. The data are unavailable on the website as the research project is still being developed.

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
