# Peer review of "Growth, Evapotranspiration, Gas Exchange and *Chl a* Fluorescence of Ipê-Rosa Seedlings at Different Levels of Water Replacement"

_plants, 2024, doi:10.3390/plants13202850_

Round 1
Reviewer 1 Report
Comments and Suggestions for Authors
This seems to be a fairly simple, straight-forward experiment that has been correctly analyzed. The writing gives probably more details than necessary, but that is better than the reverse. Some specific comments:
Somewhere early on, maybe even in the abstract, some indication of the type of plant, herb, shrub, tree would be useful to readers.
Fig 3: it was not clear what TB and Tb indicated
Fig. 5: define LWR here
Results: Normally there should be few if any references cited in Results section, but here there are numerous. Much of them could better go into the Discussion section, and keep the Results simpler.
Table 6: please define these variables in this table, and not make readers go back to the text for definitions.
Author Response
We appreciate your willingness to review the article and your valuable comments. We have made the changes, considering the three reviewers' recommendations. All changes are highlighted in red in the text. Specifically, regarding the review comments, we present the authors' responses below.
This seems to be a fairly simple, straight-forward experiment that has been correctly analyzed. The writing gives probably more details than necessary, but that is better than the reverse. Some specific comments:
Somewhere early on, maybe even in the abstract, some indication of the type of plant, herb, shrub, tree would be useful to readers.
Answer: Additional species characteristics were inserted between lines 54 and 58.
Fig 3: it was not clear what TB and Tb indicated
Answer: In the title of Figure 3, the meanings of TB and Tb were indicated, which correspond to upper basal temperature (TB) and lower basal temperature (Tb)”; based on the analyses of Figure 3, estimates were made as basal temperatures - TB = 40.88 °C and Tb = 16.60 °C.
We emphasize that there is no information in the literature on basal temperatures and accumulated thermal sums of native forest species in the Amazon, especially on the pink ipê. Therefore, citing any scientific work that has defined these parameters is impossible. The data in Figure 3 were not collected in the present study but rather in other experiments by the research group, with this species and in different environmental cultivation conditions; these experiments allowed the development of pink ipê plants (seedlings) in different environments with distinct air temperatures throughout the seedling cycle (Monteiro, 2015 - https://www.gpambienteplanta.com/_files/ugd/e8bf42_b484fd64718b44399fa5e2a132202617.pdf), and therefore, together with the data on the morphometric variables of these experiments, the basal temperatures to be used in this article were defined.
Figure 3 does not aim to evaluate the effect of temperature on plant growth and biomass accumulation but rather to estimate the upper and lower basal temperatures for the species based on data from other experiments by the research group. Plants accumulate a daily amount of energy that can be given by the accumulated thermal sum (STA), making up the energy available for carrying out their metabolic processes within optimal temperature ranges, which in turn are above the minimum condition (lower basal temperature) and below the maximum condition (upper basal temperature) of energy (temperature) that would limit the plant's development rates. Among these rates, leaf emission can be obtained based on this accumulation of thermal energy, and, in its determination, the concept of phyllochron is used, given as the time interval between the appearance of two successive leaves on the plant's main stem, expressed in °C.day.leaf-1 [10.4236/ajps.2014.515247 and 10.1590/S0100-204X2017000500001].
We agree with the reviewer that our current study did not provide air temperature variations for the pink ipê plants. Therefore, it would not be possible to define basal temperatures with this experiment since the seedlings were subjected to water deficiency. The assumptions for defining basal temperatures and thermal sum are the absence of water deficiency, nutritional deficiency and phytosanitary problems. However, this information, in addition to being new, is part of the methodology for data analysis.
Fig. 5: define LWR here
Answer: The “LWR” stands for levels replacement water. We standardize throughout the text.
Results: Normally there should be few if any references cited in Results section, but here there are numerous. Much of them could better go into the Discussion section, and keep the Results simpler.
Answer: The manuscript's authors understand the need to adapt the presentation of results and discussion, especially regarding literature citations. Therefore, two subtopics (4.1. Growth of ipê-rosa seedlings and 4.2. Evapotranspiration, crop coefficients and water sensitivity (Ky)) were included at the beginning of the “Discussion”; in this case, the citations and references that were presented in the “Results” were inserted in these two subtopics.
Table 6: please define these variables in this table, and not make readers go back to the text for definitions.
Response: Table 6 became Table 3. The authors would like to thank the reviewer for reviewing the manuscript. Based on the reviewer's suggestions, the definition of the variables in Table 3 was inserted in the title of the Table (between lines 636 and 641); this is hoped to facilitate the reader's immediate understanding of the definition of the variables presented in this table.
Reviewer 2 Report
Comments and Suggestions for Authors
Dear authors,
Please find the attachment for reviewer's comments.

Author Response
We appreciate your willingness to review the article and your valuable comments. We have made the changes, considering the three reviewers' recommendations. All changes are highlighted in red in the text. Specifically, regarding the review comments, we present the authors' responses below.
In the manuscript titled with "Growth, evapotranspiration, gas exchange and Ch a fluorescence of ipê-rosa seedlings at different levels of water replacement", the authors investigated photosynthetic activities under various levels of water regime.
- Abstract
- A reviewer doesn’t understand what is author’s point. Please describe and suggest what are new findings from this work and their importances for agricultural practice.
Answer: As requested by the reviewer, a justification was added to the summary between lines 18 and 23, highlighting the importance of evaluating water stress conditions in the plant's initial growth.
- Materials and Methods
Based on Fig. 3, the leaf area and dry mass of seedlings were not affected by temperature factor. What this result mean and how linked to this
Answer: We understand the reviewer's doubts. Therefore, we have provided a detailed explanation on the subject in question to contribute to the reviewer's scientific understanding of the study.
We emphasize that there is no information in the literature on basal temperatures and accumulated thermal sums of native forest species in the Amazon, especially on the pink ipê. Therefore, citing any scientific work that has defined these parameters is impossible. The data in Figure 3 were not collected in the present study but rather in other experiments by the research group, with this species and in different environmental cultivation conditions; these experiments allowed the development of pink ipê plants (seedlings) in different environments with distinct air temperatures throughout the seedling cycle (Monteiro, 2015 - https://www.gpambienteplanta.com/_files/ugd/e8bf42_b484fd64718b44399fa5e2a132202617.pdf), and therefore, together with the data on the morphometric variables of these experiments, the basal temperatures to be used in this article were defined.
Figure 3 does not aim to evaluate the effect of temperature on plant growth and biomass accumulation but rather to estimate the upper and lower basal temperatures for the species based on data from other experiments by the research group. Plants accumulate a daily amount of energy that can be given by the accumulated thermal sum (STA), making up the energy available for carrying out their metabolic processes within optimal temperature ranges, which in turn are above the minimum condition (lower basal temperature) and below the maximum condition (upper basal temperature) of energy (temperature) that would limit the plant's development rates. Among these rates, leaf emission can be obtained based on this accumulation of thermal energy, and, in its determination, the concept of phyllochron is used, given as the time interval between the appearance of two successive leaves on the plant's main stem, expressed in °C.day.leaf-1 [10.4236/ajps.2014.515247 and 10.1590/S0100-204X2017000500001].
We agree with the reviewer that our current study did not provide air temperature variations for the pink ipê plants. Therefore, it would not be possible to define basal temperatures with this experiment since the seedlings were subjected to water deficiency. The assumptions for defining basal temperatures and thermal sum are the absence of water deficiency, nutritional deficiency and phytosanitary problems. However, this information, in addition to being new, is part of the methodology for data analysis.
- Results and Discussion
In order to compare the difference among treatments (Figure 5), please insert the statistical results
Answer: The adjusted regressions were significant for each treatment and already incorporate the effect of time after transplanting (DAT); the last growth assessment was destructive and allowed the evaluation of the final cumulative effects between treatments (Figure 6). In this sense, the authors understand that statistical comparisons were made between treatments.
Combine Fig. 6 and Fig.
Answer: As requested by the reviewer, figures 6 and 7 have been combined to facilitate understanding of the results.
Change Table 5 to
Answer: The authors understand that presenting the numerical values of the averages in the tables becomes more interesting for future citations since studies with forest species are less common.
Table 6 and 8 don’t provide comprehensive Please remake the table showing correlation between each factor from all treatments.
Answer: Due to the normality of the data, Pearson correlation analyses were performed considering the water replacement levels. If we adopted a comprehensive correlation (with all treatments together), the morphometric variables of plant growth (Figure 6) could influence the gas exchange variables. In this case, the correlations by water replacement level allow us to understand what will happen to plants in different field conditions (practices); the seedlings subjected to 100% water replacement were in a condition of potential evapotranspiration (without water restriction) and, therefore, serve as a reference; in the other water conditions, the plants were under real evapotranspiration and in the field, when not subjected to irrigation, water replacement is dependent only on natural rainfall. In this context, the correlations generated by treatment (LWR) allow us to understand the probable responses/interactions of the gas exchange of seedlings in real growing conditions.
To facilitate reading of the results presented in Tables 6 and 8, the cells with the significant mean values at 5 and 1% were identified in red and green colors, respectively.
A reviewer recommends that Fig. 8 and 10 and Table 2 and 8 go to the supplementary data set.
Answer: The authors understand that Figure 10 and Table 2 can be designated for the supplementary data set; the other results are important for immediate understanding of the text.
Reviewer 3 Report
Comments and Suggestions for Authors
See attached

Minor editor for english language required
Author Response
We appreciate your willingness to review the article and your valuable comments. We have made the changes, considering the three reviewers' recommendations. All changes are highlighted in red in the text. Specifically, regarding the review comments, we present the authors' responses below.
Line 2: Chl a Fluorescence
Answer: It has been corrected – “Ch a” to “Chl a”
Line 38: Keyword should not be included in the title of the publication. Remove Chl a Fluorescence
Answer: The keyword “Chl a Fluorescence” was deleted and three that are not present in the title were added: “photochemical stress ; native tree”; “irrigation management”.
Line 154: References must be numbered (see also lines 160, 224, 238-239, 291-292, 328, 372, 390, 391, 400, 403, 463, 545, 571, 612, 627, 703, 734, 737, 740, 1044, 1094, 1105, 1126, 1204, 1215, 1220).
Response: The authors are grateful for the reviewer's comment. The citations were checked and kept as they are, as the citation format complies with MDPI standards.
Lines from 106 to 118: Insert a paragraph entitled: 2.1 experimental site or similar.
Answer: As requested by the reviewer, a subtopic (Line 111: 2.1. Experimental site) was added at the beginning of the materials and methods, and the numbering of the other subtopics was corrected.
Line 169: Insert specific references to instrumentation (Xxx Company, City, Country). See also lines 191-197, 200, 210, 258-259, 262-263, 268, 283, 363-365, 403, 405.
Answer: The indications were made as requested by the reviewer.
Line 401: Data analysis. This part was too much long.
Response: The authors would like to thank the reviewer for his pertinent comment regarding the data analysis. The authors verified the possibilities for reduction, however, as there is a need to specify some data collections and analyses since they were not carried out on coincident dates, and which had not yet been detailed in the methodology subtopics.
Figure 6: insert error bar.
Answer: The error bar has been inserted in Figure 6.
Line 766: Review paragraph numbering…3.4 and not 3.3.
Answer: Subtopic 3.3 has been corrected to 3.4.
Comments on the Quality of English Language
Minor editor for english language required
Response: The authors are grateful for the detailed review and indicate that after the changes were made, the article was reviewed by a native English speaker.